# Norepinephrine is required to promote wakefulness and for hypocretin-induced arousal in zebrafish

**Chanpreet Singh[†], Grigorios Oikonomou[†], David A Prober\***

Division of Biology and Biological Engineering, California Institute of Technology, Pasadena, United States

**Abstract** Pharmacological studies in mammals suggest that norepinephrine (NE) plays an important role in promoting arousal. However, the role of endogenous NE is unclear, with contradicting reports concerning the sleep phenotypes of mice lacking NE due to mutation of *dopamine β-hydroxylase* (*dbh*). To investigate NE function in an alternative vertebrate model, we generated *dbh* mutant zebrafish. In contrast to mice, these animals exhibit dramatically increased sleep. Surprisingly, despite an increase in sleep, *dbh* mutant zebrafish have a reduced arousal threshold. These phenotypes are also observed in zebrafish treated with small molecules that inhibit NE signaling, suggesting that they are caused by the lack of NE. Using genetic overexpression of hypocretin (Hcrt) and optogenetic activation of *hcrt*-expressing neurons, we also find that NE is important for Hcrt-induced arousal. These results establish a role for endogenous NE in promoting arousal and indicate that NE is a critical downstream effector of Hcrt neurons.

**\*For correspondence:** dprober@caltech.edu

**[†]**These authors contributed equally to this work

**Competing interests:** The authors declare that no competing interests exist.

## Introduction

Sleep remains among the most persistent and perplexing mysteries in modern biology. Several studies have shown that neuronal centers that regulate sleep and wakefulness lie predominantly in the hypothalamus and brainstem (*Pace-Schott and Hobson, 2002*; *Saper et al., 2005*), and many of the neurotransmitters and neuropeptides employed by these centers are known. However, it remains unclear how these centers interact with each other, and what specific functions are fulfilled by the neurotransmitters and neuropeptides they employ. Norepinephrine (NE) is one of the most abundant neurotransmitters in the central and peripheral nervous systems, and has been implicated in many aspects of physiology and behavior, including cognition, attention, reward, locomotion and arousal (*Berridge and Waterhouse, 2003*; *Weinshenker and Schroeder, 2007*; *Sara, 2009*). Small molecule activators of NE signaling have been shown to increase wakefulness, whereas inhibitors promote sleep (*Berridge et al., 2012*). These results suggest that NE plays a significant role in promoting arousal. However, the role of endogenous NE in regulating the sleep/wake cycles of vertebrates remains unclear, with contradicting reports concerning the sleep phenotype of mice that do not synthesize NE due to mutation of *dopamine beta hydroxylase* (*dbh*) (*Hunsley and Palmiter, 2003*; *Ouyang et al., 2004*).

The locus coeruleus (LC) in the brainstem is a major arousal promoting center and a major source of NE in the brain (reviewed in *Berridge and Waterhouse, 2003*). Optogenetic studies in mice have shown that activation of the LC promotes sleep-to-wake transitions, while inhibition of the LC reduces time spent awake (*Carter et al., 2010*, *2012*). These studies also showed that activation of the LC plays an important role in mediating the arousing effects of *hypocretin* (*hcrt*)-expressing neurons (*Carter et al., 2010*, *2012*), which constitute an important arousal center in the hypothalamus (*Sutcliffe and de Lecea, 2002*). These observations suggest that NE synthesized in the LC could be

**eLife digest** Although the neural circuits that regulate sleep and wakefulness have yet to be fully identified, the importance of at least two brain regions is well established. These are the hypothalamus, a structure deep within the brain that controls a number of basic activities including hunger, thirst and sleep; and the brainstem, which connects the brain with the spinal cord.

Specific neurons within the hypothalamus and brainstem regulate the sleep–wake cycle by signaling to one another using chemicals called neurotransmitters and neuropeptides. Throughout the day, some hypothalamic neurons release a neuropeptide called hypocretin, which helps maintain wakefulness. Hypocretin acts on neurons within the brainstem and causes them to release other neurotransmitters that promote wakefulness. However, the identity of these molecules is unclear.

One candidate is norepinephrine. Drugs that enhance the effects of norepinephrine increase wakefulness, whereas those that block norepinephrine signaling promote sleep. Despite this, mice that have been genetically modified to lack the enzyme that produces norepinephrine exhibit relatively normal sleep. This may be because in mammals, norepinephrine also has important roles outside the brain, thus complicating the effects of this genetic modification on behavior. Alternatively, while zebrafish that lack norepinephrine are healthy, mice containing this modification die early in development. Treating these mice with a specific drug allows them to survive, but might affect their behavior.

To clarify the role of norepinephrine and its interaction with hypocretin, Singh, Oikonomou and Prober created a new animal model by genetically modifying zebrafish. In contrast to mice, zebrafish that were unable to make norepinephrine slept more than normal fish, although they were also lighter sleepers and were more prone to being startled. A genetic modification that increases hypocretin signaling induces insomnia; Singh, Oikonomou and Prober found that this occurs only in animals with normal levels of norepinephrine. Thus, these experiments indicate that hypocretin does indeed promote wakefulness though norepinephrine.

The work of Singh, Oikonomou and Prober has clarified the role of norepinephrine in regulating the sleep–wake cycle. These findings could help in the development of drugs that target the neurons that make hypocretin, which may improve treatments for sleep disorders.

involved in mediating Hcrt-induced arousal. However, in addition to NE, the LC also produces other neurotransmitters and neuropeptides that affect sleep, including dopamine (*Dzirasa et al., 2006*), neuropeptide Y (*Dyzma et al., 2010*), neurotensin (*Erwin and Radcliffe, 1993*) and vasopressin (*Born et al., 1992*). Manipulations of the LC likely affect the synaptic levels of these other neurotransmitters and neuropeptides. Therefore, it is unclear whether NE is required to mediate the arousing effects of LC neurons and Hcrt signaling, or if this is accomplished by another factor in LC neurons.

To address the role of NE in regulating vertebrate sleep/wake states, we generated *dbh* mutant zebrafish that do not produce NE. Contrary to *dbh* null mice (*Thomas et al., 1995*), zebrafish *dbh* mutants develop normally and are viable. Importantly, they also exhibit a dramatic increase in sleep. Interestingly, despite increased sleep, these animals display a reduced arousal threshold. Using this mutant, we show that NE is important for arousal that is induced by either genetic overexpression of the Hcrt neuropeptide or optogenetic activation of Hcrt neurons. These results clarify the role of NE in regulating vertebrate sleep and establish a role for NE in mediating Hcrt-induced wakefulness.

## Results

### Pharmacological inhibition of NE signaling increases sleep

Previous pharmacological studies in zebrafish (*Rihel et al., 2010*) suggested that the noradrenergic system is an important regulator of sleep and wakefulness in zebrafish, similar to mammals (*Berridge et al., 2012*). Therefore, we reasoned that the zebrafish model system could provide a new platform for exploring the role of endogenous NE in sleep. Three classes of receptors mediate NE signaling: the activating alpha1-and beta-adrenergic receptors and the inhibitory alpha2 adrenergic receptors. For each class there exist at least 5 paralogs in the zebrafish genome, making pharmacological manipulations more practical than genetic manipulations. We first tested the effects of prazosin,

a well-established alpha1-adrenergic receptor inhibitor, on zebrafish behavior. We found that, compared to larvae exposed to dimethyl sulfoxide (DMSO) vehicle alone, larvae exposed to 100 µM prazosin showed lower overall activity (*Figure 1A,D*) and activity when awake (*Figure 1B,E*) during both day and night, as well as an increase in sleep during both day (+140%) and night (+60%) (*Figure 1C,F*; *Figure 1—figure supplement 1*). This increase in sleep was primarily due to an increase in the number of sleep bouts (+110% during the day and +70% during the night, *Figure 1G*), as well as a smaller increase in sleep bout duration (+20% during the day) (*Figure 1H*). We also observed a 23% reduction in sleep latency at night in prazosin-treated animals (*Figure 1I*).

We further investigated whether treatment with the alpha2-adrenergic receptor agonist clonidine or the beta1-adrenergic receptor antagonist bopindolol affects the sleep of larval zebrafish. We found that the main effect of clonidine treatment was a reduction in day activity (−48%) and increase in day sleep (+90%), with no effect in night activity or sleep (*Figure 1—figure supplement 2*), suggesting a day-specific role for alpha2 receptors in sleep regulation. Bopindolol treatment resulted in reduction of night activity (−40%) and increase in both day (+82%) and night (+60%) sleep (*Figure 1—figure supplement 3*).

## *dbh* mutant larvae exhibit increased sleep

Encouraged by our pharmacological studies, we decided to investigate the role of endogenous NE in sleep regulation using genetics. To this end we used the zinc-finger nuclease approach (*Foley et al., 2009*), a well established technique in zebrafish that induces targeted mutations with few off-target lesions (*Doyon et al., 2008*; *Meng et al., 2008*; *Gupta et al., 2011*), to generate zebrafish containing a mutation in the single zebrafish *dopamine beta-hydroxylase* (*dbh*) gene (*Chen et al., 2013a*), which encodes the enzyme that converts dopamine to NE. The *dbh* mutant allele that we generated contains a four nucleotide insertion in the third exon, which gives rise to a premature stop codon, resulting in a predicted 178 amino acid peptide lacking the DBH active site, instead of the full length 614 amino acid protein (*Figure 2—figure supplement 1A*).

Although the murine *dbh* knockout (*Thomas et al., 1995*) displays no overt developmental defects, only 10% of *dbh−/−* pups produced by *dbh+/−* mothers survive embryonic development and no *dbh−/−* pups are born to *dbh−/−* mothers. In contrast to the mouse knockout, female *dbh−/−* zebrafish produced a normal number of *dbh−/−* larvae when mated to *dbh+/−* males (out of 287 larvae genotyped from 3 independent crosses, 49% were *dbh+/−* and 51% were *dbh−/−*). We verified that *dbh−/−* larvae completely lack NE using an ELISA assay (*Figure 2—figure supplement 1B*). Using in situ hybridization (ISH) we also found that the level of *dbh* mRNA in *dbh−/−* larvae is much lower than in their *dbh+/−* siblings, presumably due to nonsense mediated decay of the mutant transcript (*Isken and Maquat, 2007*) (*Figure 2—figure supplement 1C*). While only 40% of *dbh* mutant mice that survive embryogenesis reach adulthood (*Thomas et al., 1995*), *dbh−/−* zebrafish embryos developed to adulthood normally (24/96 (25%) of embryos produced by a male *dbh+/−* crossed to female *dbh+/−*, and raised to adulthood, were identified as *dbh−/−*).

We used the *dbh* mutant to determine whether genetic ablation of NE affects the sleep/wake patterns of zebrafish larvae. We observed that *dbh* mutants display lower overall activity (*Figure 2A,D*) and activity when awake (*Figure 2B,E*) during both day and night. Importantly, we found that *dbh−/−* larvae sleep much more during both day (+185%) and night (+57%) than sibling controls (*Figure 2C,F*). During the day, the increase in sleep was due solely to more frequent sleep bouts (+220%), while during the night both longer (+17%) and more frequent (+25%) sleep bouts occurred (*Figure 2G,H*). Furthermore, sleep latency at night in the mutants was reduced by 40% (*Figure 2I*). It is interesting to note that, as expected, treatment of *dbh* mutants with 100 µM prazosin did not affect sleep/wake behavior compared to DMSO controls (*Figure 2—figure supplement 2*), suggesting that prazosin-induced behavioral effects are specific to NE signaling.

## *dbh* mutant larvae exhibit lower arousal threshold

Since *dbh* mutants exhibit increased sleep, we decided to investigate whether their arousal threshold is also altered. To this end we delivered mechano-acoustic tapping stimuli of variable intensities to larvae while monitoring their behavior using a videotracking system. We delivered a stimulus once a minute during the night, and determined the fraction of larvae that responded to each stimulus. This dataset allowed us to construct dose–response curves (*Figure 3A*) and calculate the tapping intensity at which 50% of larvae responded to the stimulus (effective tap power 50, $ETP_{50}$). Surprisingly, the $ETP_{50}$ for *dbh−/−* larvae was 2.1 compared to 3.1 for their *dbh+/+* siblings

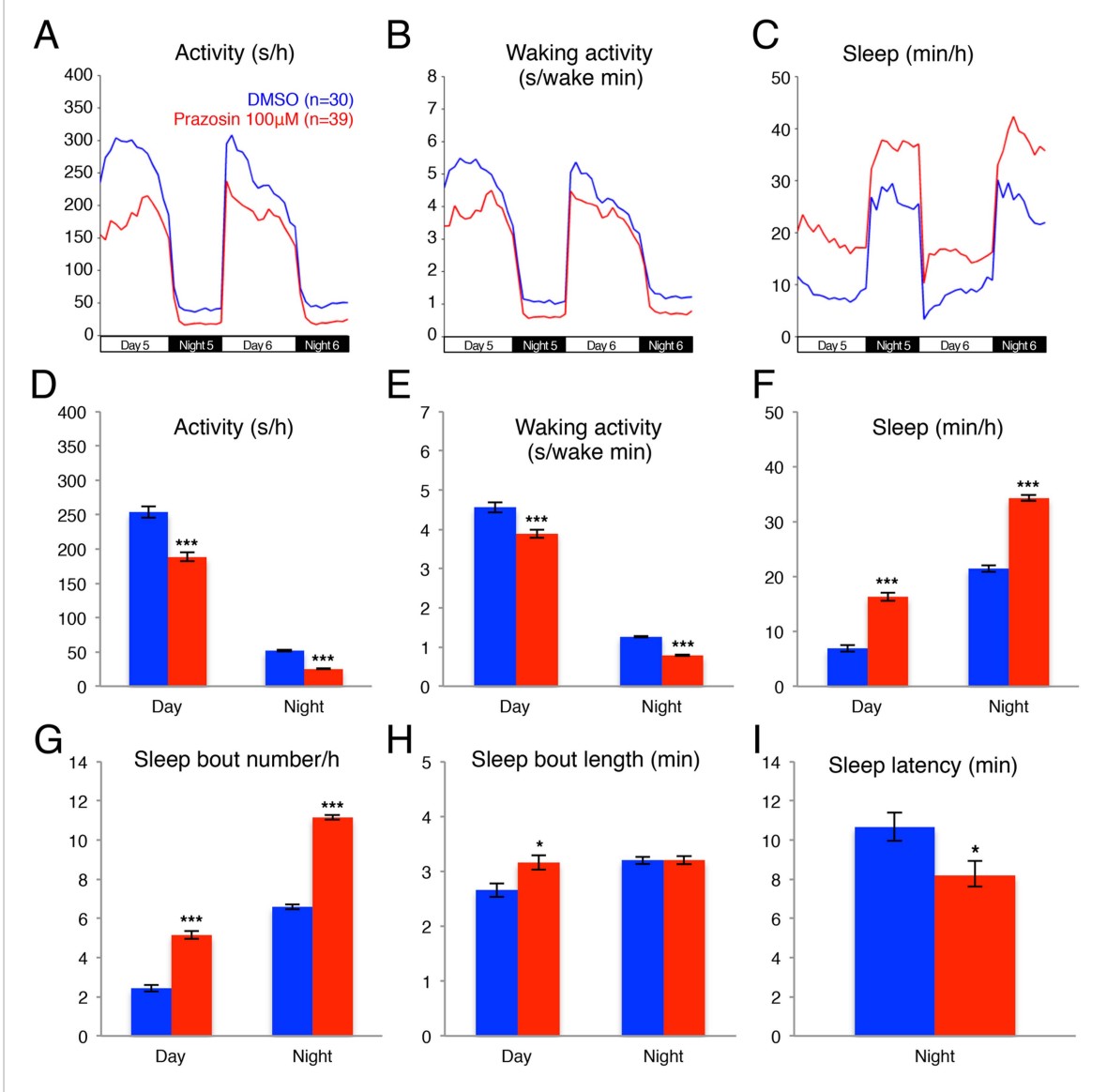

**Figure 1**. Prazosin treated larvae are less active and sleep more than vehicle treated controls. Representative activity (**A**), waking activity (amount of locomotor activity while awake) (**B**) and sleep (**C**) traces of vehicle (blue) and prazosin (red) treated zebrafish larvae. Bar graphs of activity (**D**), waking activity (**E**), sleep (**F**), sleep bout number (**G**), sleep bout length (**H**) and sleep latency (**I**) from three combined experiments (n > 180 for each condition). Bars represent mean ± s.e.m. *, p < 0.05 and ***, p < 0.0001 by one-way ANOVA.

The following figure supplements are available for figure 1:

**Figure supplement 1**. Prazosin night sleep dose–response curve.

**Figure supplement 2**. Clonidine treated larvae are less active and sleep more than vehicle treated controls during the day.

**Figure supplement 3**. Bopindolol treated larvae sleep more than vehicle treated controls.

(32% decrease) (p < 0.0001 by extra sum-of-squares F test), suggesting that although the *dbh* mutants sleep more, they have a reduced arousal threshold. In addition to a lower $ETP_{50}$, *dbh*−/− larvae also exhibited an increased maximal response fraction, from 0.43 for *dbh*−/− compared to 0.34 for *dbh*+/+ (p < 0.0001 by extra sum-of-squares F test). Interestingly, treatment with the alpha1-adrenergic

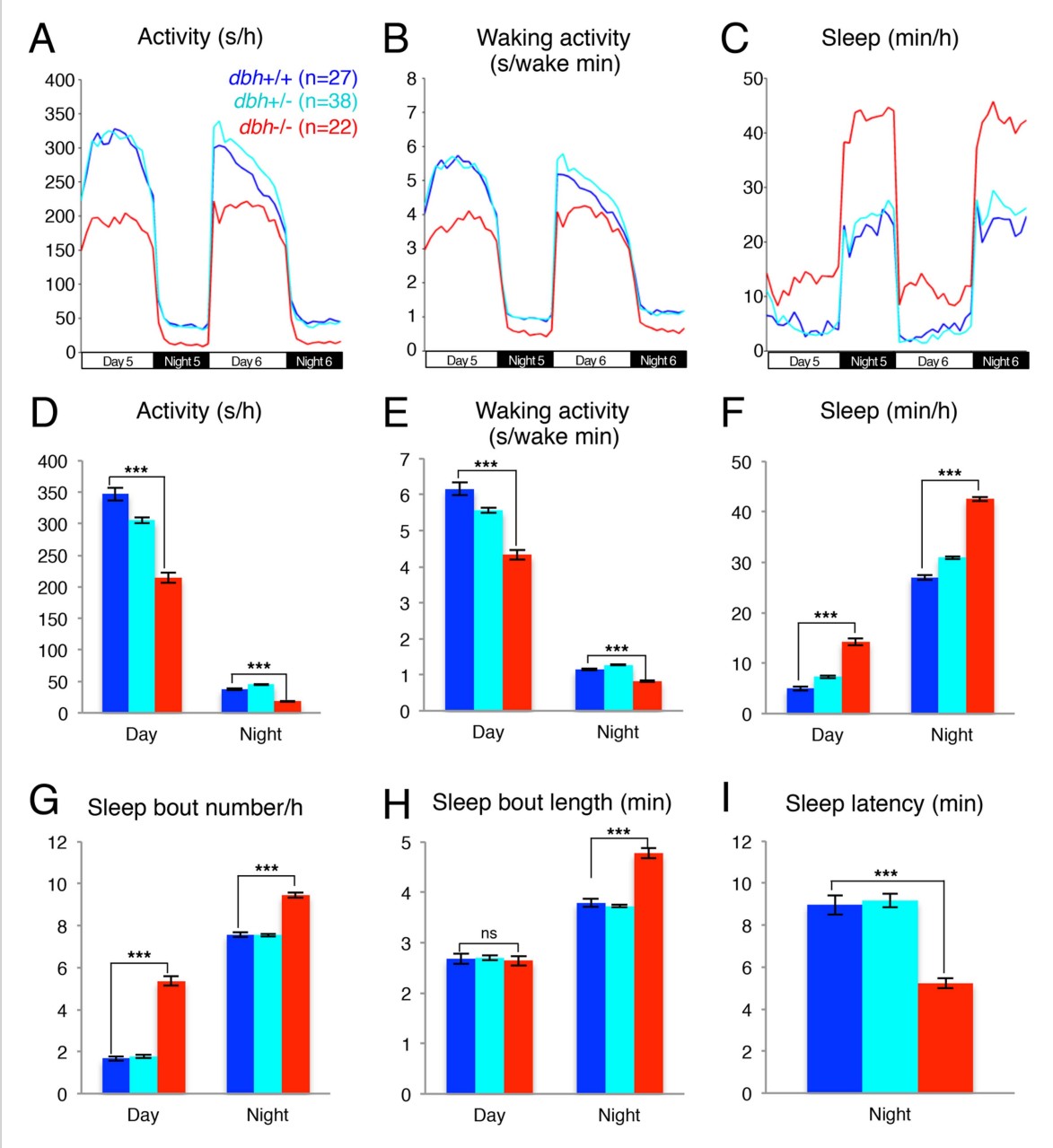

**Figure 2**. *dbh* mutant larvae are less active and sleep more than sibling controls. Representative activity (**A**), waking activity (**B**) and sleep (**C**) traces of *dbh+/+* (blue) *dbh+/−* (cyan) and *dbh−/−* (red) zebrafish larvae. Bar graphs show mean ± s.e.m. activity (**D**), waking activity (**E**), sleep (**F**), sleep bout number (**G**), sleep bout length (**H**) and sleep latency (**I**) from seven combined experiments (n > 250 for each genotype). n indicates number of larvae. ***, p < 0.0001 and ns, not significant by one-way ANOVA.

The following figure supplements are available for figure 2:

**Figure supplement 1**. Verification of zebrafish *dbh* mutant.

**Figure supplement 2**. Treatment of *dbh−/−* fish with prazosin does not alter sleep/wake behavior.

antagonist prazosin or the alpha2-adrenergic agonist clonidine did not affect the arousal threshold of zebrafish larvae (data not shown). However, larvae treated with the alpha2-adrenergic antagonist bopindolol showed a 54% decrease in their $ETP_{50}$ from 5.7 to 2.6 (*Figure 3B*, p < 0.0001 by extra

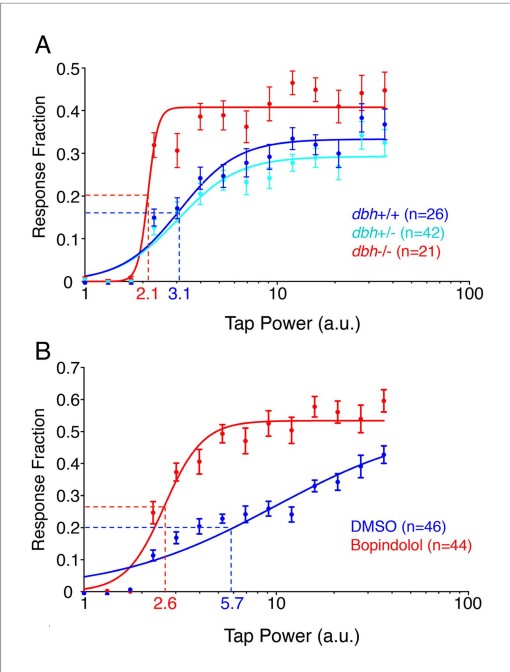

**Figure 3.** *dbh* mutants and bopindolol-treated WT larave exhibit lower arousal threshold. Stimulus-response curves generated by a tapping assay for *dbh*–/– and sibling control larvae (**A**), and bopindolol and DMSO treated WT larvae (**B**). Thirty trials were performed at each stimulus intensity, with a 1 min inter-trial interval. Each data point indicates mean ± s.e.m. Stimulus-response curves were constructed using the non-linear variable slope module in Prism and fitted using ordinary least squares. Dashed lines mark the $ETP_{50}$ values for different genotypes and drug treatments. *dbh*–/– have an $ETP_{50}$ value of 2.1 vs 3.1 for sibling controls (32% decrease, $p < 0.0001$ by extra sum-of-squares F test) (**A**), and bopindolol treated animals have an $ETP_{50}$ value of 2.6 vs 5.7 for sibling controls (54% decrease, $p < 0.0001$ by extra sum-of-squares F test) (**B**). n indicates number of larvae.

sum-of-squares F test). The maximal response fraction was increased from 0.42 to 0.53 but this change did not achieve statistical significance ($p = 0.08$).

## Hypocretin (hcrt) overexpression-induced wakefulness requires NE

The Hcrt neuropeptide has been shown to promote wakefulness and inhibit sleep in zebrafish larvae (*Prober et al., 2006*) and mammals (reviewed in *Alexandre et al., 2013*; *Sakurai, 2013*), and stimulation of Hcrt neurons has been shown to promote sleep to wake transitions in mice (*Adamantidis et al., 2007*). Hcrt has been proposed to promote wakefulness in part by stimulating the noradrenergic LC based on several lines of evidence. First, the LC is densely innervated by Hcrt neurons in both zebrafish (*Prober et al., 2006*) (*Figure 4—figure supplement 1A,B*) and mammals (*Peyron et al., 1998*; *Chemelli et al., 1999*; *Date et al., 1999*; *Horvath et al., 1999*) and LC neurons express the Hcrt receptor (*Horvath et al., 1999*; *Bourgin et al., 2000*; *Prober et al., 2006*). It is worth noting, however, that the medulla oblongata is another source of NE in the brain, and it also receives Hcrt projections in zebrafish (*Figure 4—figure supplement 1C*) and mammals (*Ciriello et al., 2003*; *Zhang et al., 2004*). Second, application of Hcrt peptide depolarizes LC neurons in brain slices (*Hagan et al., 1999*) and in vivo (*Bourgin et al., 2000*). Third, acute inhibition of LC neurons using halorhodopsin inhibits the arousing effects of stimulating Hcrt neurons (*Carter et al., 2012*). However, despite the evidence that the LC mediates Hcrt-induced arousal, it is unknown whether this process requires NE.

The mammalian genome contains two *hcrt receptor* (*hcrtr*) paralogs, both of which are required for the increase in locomotor activity and decrease in sleep observed after injection of Hcrt peptide (reviewed in *Alexandre et al., 2013*; *Sakurai, 2013*). To determine whether the single zebrafish *hcrtr* ortholog is required for the Hcrt overexpression phenotype in zebrafish larvae, we mated *Tg(hsp:Hcrt)* zebrafish to a previously described *hcrtr* mutant (*Yokogawa et al., 2007*). We subjected the larval progeny to heat shock (HS) during the sixth day post fertilization (dpf) to induce Hcrt overexpression (*Prober et al., 2006*). We then compared the amount of sleep on night 6 (post-HS) to night 5 (pre-HS). We previously showed that Hcrt protein levels remain elevated for over 48 hr after HS, with no detectable decrease during the first 24 hr after HS (*Prober et al., 2006*), so overexpressed Hcrt protein should remain at high levels throughout the night following HS. *hcrtr+/–* and *hcrtr–/–* larvae reacted similarly to the HS, with no significant difference in the amount of sleep between the two genotypes (*Figure 4—figure supplement 2A,B*). Hcrt overexpression decreased the sleep of *Tg(hsp:Hcrt);hcrtr+/–* larvae by 50% (*Figure 4—figure supplement 2C,D*), as previously described for *Tg(hsp:Hcrt);hcrtr+/+* larvae (*Prober et al., 2006*). However, no decrease in sleep was observed in *Tg(hsp:Hcrt);hcrtr–/–* larvae following HS (*Figure 4—figure supplement 2C,D*). These results indicate that the decrease in sleep following overexpression of Hcrt requires a functional *hcrtr*.

To test the hypothesis that NE is required for Hcrt-induced wakefulness, we mated *Tg(hsp:Hcrt)+/−; dbh+/−* fish to *dbh−/−* fish and heat-shocked the larval progeny during day 6. We first asked whether the presence or absence of NE affects the response of larvae to HS. Larvae lacking the *hsp:Hcrt* transgene were similarly affected by the HS independently of whether they produced NE or not (no significant difference observed between *dbh+/−* and *dbh−/−* larvae, **Figure 4A,B**). We therefore conclude that lack of NE does not affect the response of larvae to HS. We next asked whether NE is required for Hcrt overexpression-induced arousal (**Figure 4C,D**). As expected, the amount of sleep exhibited by *Tg(hsp:Hcrt)* larvae was reduced following Hcrt overexpression (**Figure 4D**). Specifically, the sleep of *Tg(hsp:Hcrt)+/−;dbh+/−* larvae was reduced to 40% of the pre-HS value (**Figure 4D**). However, the sleep of *Tg(hsp:Hcrt)+/−;dbh−/−* larvae was reduced to 75% of the pre-HS value (**Figure 4D**). Thus, the majority of Hcrt-induced sleep loss is blocked in the absence of NE. Importantly, treatment of *Tg(hsp:Hcrt)+/−* animals with prazosin similarly inhibits Hcrt overexpression-induced arousal (**Figure 4—figure supplement 3**) suggesting that the *dbh* mutant phenotype is indeed due to the blocking of noradrenergic signaling. We also confirmed that the suppressed effect of Hcrt overexpression on sleep in *dbh* and *hcrtr* mutants is not due to an effect of the mutations on

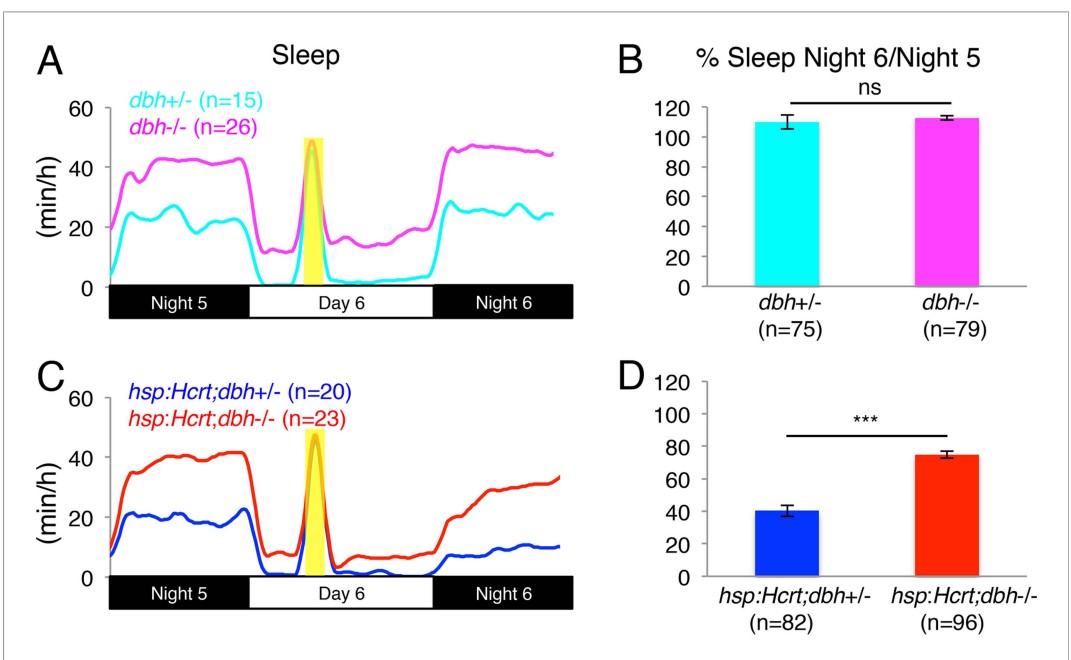

**Figure 4**. Reduced sleep at night due to Hcrt overexpression is suppressed in *dbh* mutants. (**A**, **C**) Sleep traces from a single representative experiment; yellow region indicates time of heat shock (HS) during day 6. (**B**, **D**) Bar graphs show mean ± s.e.m. percentage change in sleep during night 6 compared to night 5 from four combined experiments. (**B**) In the absence of the *hsp:Hcrt* transgene, HS during day 6 has no significant effect on *dbh−/−* larvae compared to *dbh+/−* larvae. (**D**) The reduced sleep induced by Hcrt overexpression is significantly diminished in *dbh−/−* larvae compared to *dbh+/−* sibling controls. n indicates number of larvae. ***, p < 0.0001 and ns, not significant by one-way ANOVA.

The following figure supplements are available for figure 4:

**Figure supplement 1**. Hcrt neurons project to *dbh* expressing cells.

**Figure supplement 2**. Reduced sleep at night following Hcrt overexpression requires the *hcrtr*.

**Figure supplement 3**. Reduced sleep at night due to Hcrt overexpression is suppressed by prazosin.

**Figure supplement 4**. The heat shock promoter is not suppressed in *hcrtr* or *dbh* mutants.

the efficacy of the HS overexpression system (*Figure 4—figure supplement 4*). These results indicate that NE plays an important role in Hcrt-mediated arousal, but suggest that Hcrt also inhibits sleep via NE-independent mechanisms.

## Activation of hcrt neurons using ChR2 increases locomotor activity

As an alternative approach to study the effects of activated Hcrt signaling on larval zebrafish behavior, we generated transgenic zebrafish in which the *hcrt* promoter (*Faraco et al., 2006*) drives expression of channelrhodopsin-2 fused to EYFP (*hcrt:ChR2-EYFP*). Using immunofluorescence, we verified that ChR2-EYFP is exclusively expressed in Hcrt neurons, and that all Hcrt neurons express ChR2-EYFP (*Figure 5—figure supplement 1A*). To test whether the *hcrt:ChR2-EYFP* transgene can activate Hcrt neurons, we exposed larvae to blue light for 30 min, and then performed double fluorescent ISH using probes specific for *hcrt* and *c-fos*. We observed that 40% of Hcrt neurons in *Tg(hcrt:ChR2-EYFP)* larvae expressed *c-fos*, compared to only 5% of Hcrt neurons in control *Tg(hcrt:EGFP)* larvae (*Figure 5A,B*). This result confirms that the light paradigm employed activates Hcrt neurons.

To test the behavioral effect of activating Hcrt neurons using ChR2, we developed a large-scale and non-invasive optogenetic behavioral assay. We modified the locomotor activity assay by adding an array of blue and red LEDs that uniformly illuminates the 96-well plate, thus allowing simultaneous light stimulation of 96 freely behaving larvae. Larvae at 5 dpf were transferred to a 96-well plate and kept in the dark for 8 hr in the behavioral chamber, after which they were exposed to either red or blue light for 30 min. The behavior of *Tg(hcrt:ChR2-EYFP)* larvae was compared to their non-transgenic siblings throughout the night with an inter-trial interval of 3 hr. A typical response to blue or red light included a burst of activity at light onset lasting approximately 30 s, followed by a return to near baseline activity levels, a gradual increase in activity that reached a plateau for the remainder of the illumination phase, and a burst of activity when the lights were turned off. The bursts of activity observed at light onset and offset were similar for transgenic and non-transgenic larvae and were excluded from behavioral analysis. *Tg(hcrt:ChR2-EYFP)* larvae and their non-transgenic siblings exhibited similar levels of locomotor activity when illuminated with red light (*Figure 5C,D*), a treatment that does not activate ChR2. In contrast, when illuminated with blue light the total locomotor activity of *Tg(hcrt:ChR2-EYFP)* larvae increased by 46% compared to non-transgenic siblings (*Figure 5E,F*). Activation of Hcrt neurons by ChR2 also altered the dynamics of locomotor activity. Specifically *Tg(hcrt:ChR2-EYFP)* larvae displayed a 25% increase in the maximum activity level reached during stimulation (*Figure 5—figure supplement 1B*) and reached this maximum activity level 20% faster (*Figure 5—figure supplement 1C*) than sibling controls.

We next asked whether the increase in locomotor activity observed upon activation of zebrafish Hcrt neurons using ChR2 requires the *hcrtr*. We first confirmed that *hcrtr−/−* larvae in the absence of the *hcrt:ChR2-EYFP* transgene respond to blue light in a manner similar to their *hcrtr+/−* and *hcrtr+/+* siblings. Indeed, we observed no significant difference in the locomotor activity of larvae of the three genotypes during exposure to blue light (*Figure 6A,B*). However, *Tg(hcrt:ChR2-EYFP);hcrtr−/−* larvae were 25% less active than their *Tg(hcrt:ChR2-EYFP);hcrtr+/−* and *Tg(hcrt:ChR2-EYFP);hcrtr+/+* siblings in response to blue light. This result indicates that a functional *hcrtr* is important for Hcrt neuron-induced arousal (*Figure 6C,D*).

Mammalian and zebrafish Hcrt neurons coexpress other neurotransmitters and neuropeptides, such as glutamate and dynorphin (*Chou et al., 2001*; *Rosin et al., 2003*; *Appelbaum et al., 2009*; *Liu et al., 2015*), which could play a role in locomotor activity that is induced by stimulation of Hcrt neurons. Determining whether this is the case requires directly comparing locomotor activity in *hcrtr−/−* larvae with and without the *hcrt:ChR2-EYFP* transgene. Since these larvae were previously tested in separate experiments, we repeated the experiment but included all six genotypes in the same behavioral plates. As expected, blue light stimulation significantly increased locomotor activity for *Tg (hcrt:ChR2-EYFP);hcrtr+/+* compared to *hcrtr+/+* larvae, and for *Tg(hcrt:ChR2-EYFP);hcrtr+/−* compared to *hcrtr+/−* larvae (*Figure 6—figure supplement 1*). *Tg(hcrt:ChR2-EYFP);hcrtr−/−* were also more active than *hcrtr−/−* larvae, although the difference was much smaller than for *Tg(hcrt: ChR2-EYFP);hcrtr+/+* compared to *hcrtr+/+* larvae. Furthermore, *Tg(hcrt:ChR2-EYFP);hcrtr+/+* larvae were significantly more active than *Tg(hcrt:ChR2-EYFP);hcrtr−/−* larvae. These results suggest that most, but not all, of the effect of stimulating Hcrt neurons on locomotor activity is due to Hcrt signaling, similar to results obtained in mammals (*Adamantidis et al., 2007*).

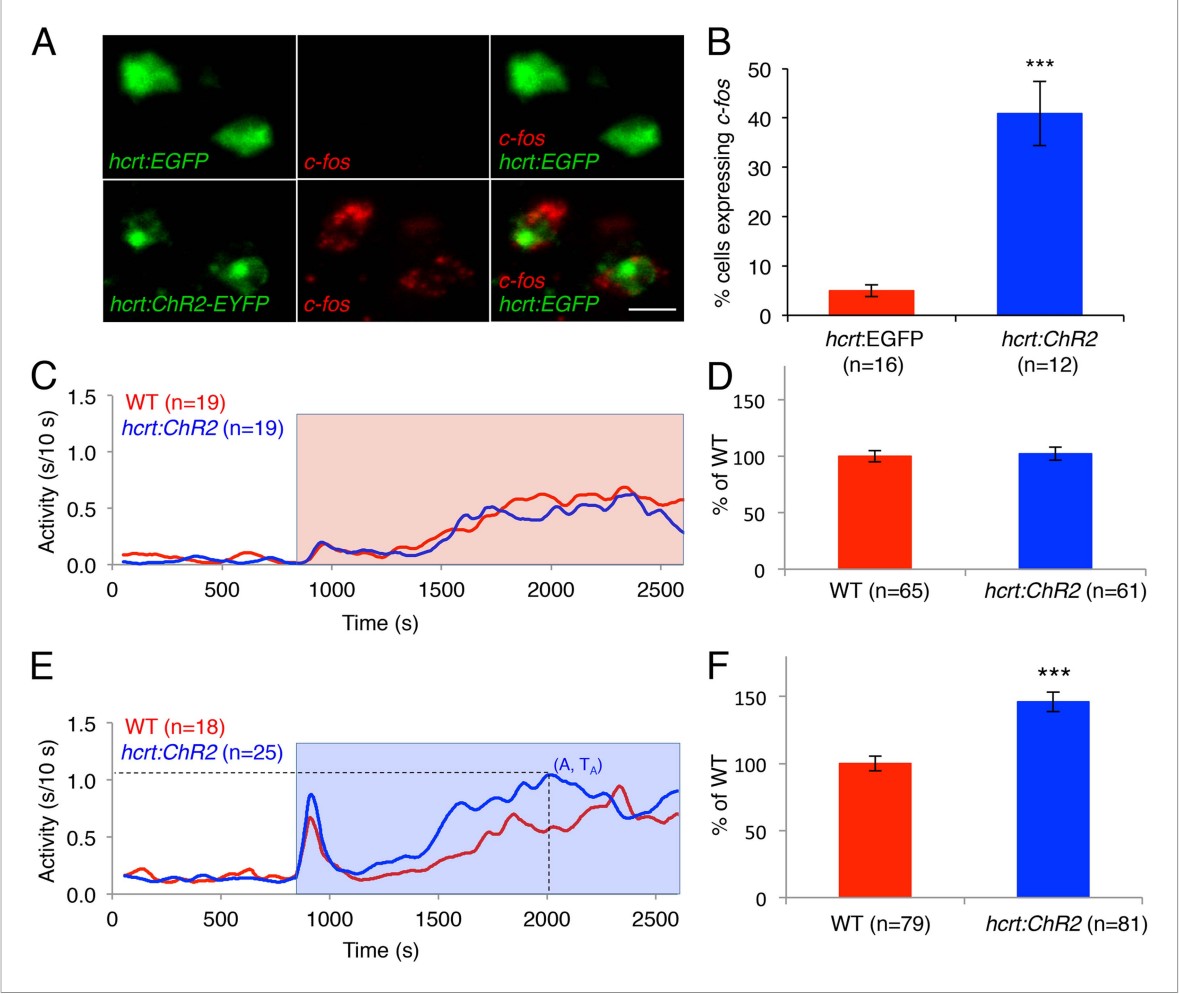

**Figure 5**. Optogenetic activation of Hcrt neurons increases locomotor activity. (**A**) Representative images of Hcrt neurons co-labeled with EYFP or EGFP (green) and *c-fos* (red). Scale bar = 10 μm. (**B**) Mean ± s.e.m. percentage of EYFP- or EGFP-expressing neurons that also express *c-fos* in *Tg(hcrt:EGFP)* and *Tg(hcrt:ChR2-EYFP)* larvae. (**C**) Representative locomotor activity trace during red light exposure. (**D**) Average locomotor activity relative to WT siblings during red light exposure. Data is pooled from 3 experiments and is represented as mean ± s.e.m. (**E**) Representative locomotor activity trace during blue light exposure. (**F**) Average locomotor activity relative to WT siblings during blue light exposure. Data is pooled from 4 experiments and is represented as mean ± s.e.m. Red and blue boxes in (**C**) and (**E**) indicate periods of red and blue light illumination. 'A' is the maximum activity reached for larvae of a particular genotype and '$T_A$' is the time taken to reach maximum activity A. n indicates number of larvae. ***, p < 0.0001 by one-way ANOVA.

The following figure supplement is available for figure 5:

**Figure supplement 1**. Specific expression of ChR2-EYFP in Hcrt neurons and temporal dynamics of *hcrt:ChR2-EYFP* induced locomotor activity.

Thus, similar to Hcrt overexpression (*Prober et al., 2006*), optogenetic activation of only ~8 larval zebrafish Hcrt neurons (an average of 40% of the 20 Hcrt neurons are *c-fos* positive upon stimulation with ChR2; *Figure 5B*) increases locomotor activity, consistent with a role for Hcrt neurons in promoting arousal (*Lee et al., 2005*; *Mileykovskiy et al., 2005*; *Adamantidis et al., 2007*; *Carter et al., 2012*). This effect appears to be significantly stronger than the phenotype described in mice (*Adamantidis et al., 2007*; *Carter et al., 2012*), suggesting that zebrafish may be more sensitive to the arousing effect of Hcrt. These experiments also demonstrate for the first time a large-scale and non-invasive application of optogenetics to manipulate the activity of a small population of neurons deep in the brain of freely behaving animals. This approach may be generally useful for studies of neuronal circuit function in regulating vertebrate behaviors.

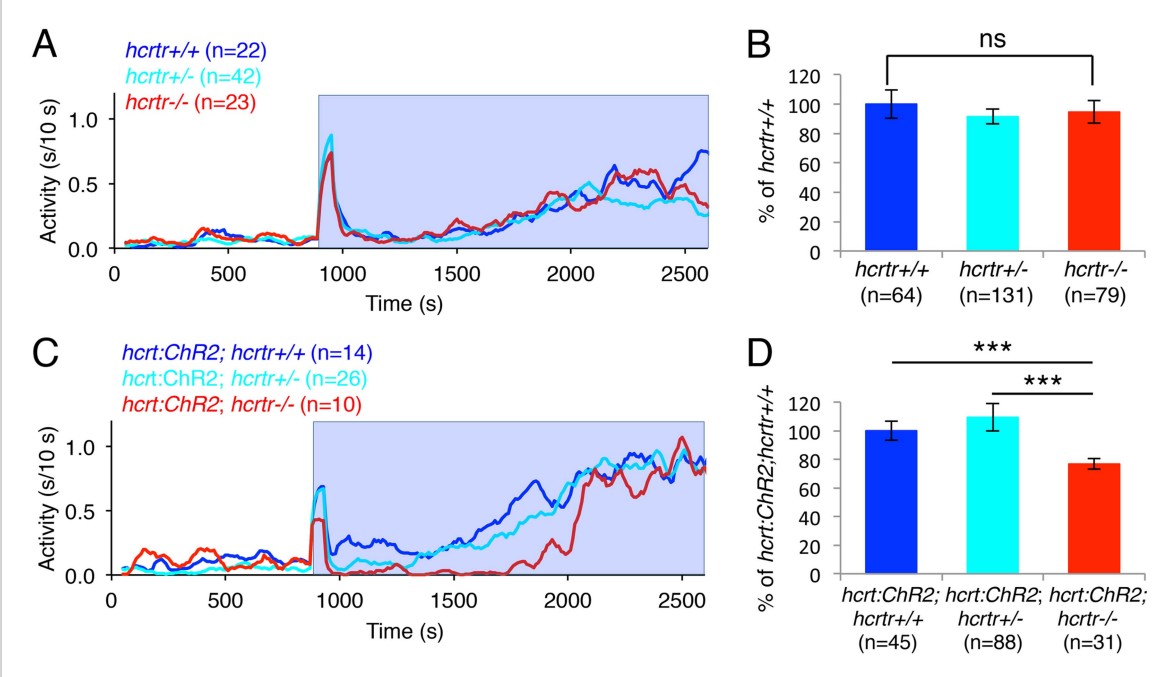

**Figure 6**. The *hcrtr* is required for Hcrt neuron-induced increased locomotor activity. (**A**, **C**) Representative locomotor activity traces for *hcrtr–/–* and sibling controls without (**A**) and with (**C**) the *hcrt:ChR2-EYFP* transgene during blue light exposure. (**B**, **D**) Average locomotor activity relative to sibling controls. Data is pooled from 3 experiments in both cases. Data is represented as mean ± s.e.m. and is plotted as percentage of *hcrtr+/+* (**B**) or *hcrt:ChR2-EYFP; hcrtr+/+* (**D**). n indicates number of larvae. ***, p < 0.0001 and ns, not significant by one-way ANOVA followed by Tukey's test to correct for multiple comparisons.

The following figure supplement is available for figure 6:

**Figure supplement 1**. Hcrt signaling is required for Hcrt neuron-induced increased locomotor activity.

## Optogenetic stimulation of hcrt neurons activates *dbh*-expressing LC neurons

Based on our observation that larval zebrafish Hcrt neurons innervate the LC (*Figure 4—figure supplement 1*) and previous studies in mammals and zebrafish (*Peyron et al., 1998*; *Chemelli et al., 1999*; *Date et al., 1999*; *Horvath et al., 1999*; *Kaslin et al., 2004*; *Prober et al., 2006*), we sought to explore the functional interaction between the two neuronal populations using a combination of optogenetics and calcium imaging. Using a transient injection approach (see 'Materials and methods'), we generated *Tg(hcrt:ChR2-EYFP)* and *Tg(hcrt:EGFP)* larvae in which single LC neurons expressed the genetically encoded calcium indicator GCaMP6s. After paralyzing and mounting these larvae in low melt agarose, we illuminated with a 488 nm laser a region of the brain containing the Hcrt neuron soma with 10 short pulses (0.3 s each over 3.2 s total), then imaged GCaMP6s fluorescence in the LC for 30 s (*Figure 7A*). We reasoned that even though we did not stimulate Hcrt neurons and image the LC simultaneously, the short time interval between the final stimulation pulse and the initiation of imaging (<0.1 s), and the relatively slow kinetics of GCaMP6s fluorescence changes (*Chen et al., 2013b*), would allow us to detect Hcrt neuron-induced effects on LC neuron activity. Indeed, over multiple trials in several animals we observed a significant increase in GCaMP6s fluorescence in *Tg(hcrt:ChR2-EYFP)* larvae, but not in *Tg(hcrt:EGFP)* larvae (*Figure 7B–E*, *Videos 1* and *2*), indicating that stimulation of Hcrt neurons results in activation of the LC in zebrafish larvae.

## NE is important for locomotor activity induced by optogenetic activation of hcrt neurons

Optogenetic studies in rodents have shown that acute inhibition of LC neurons blocks the wake-promoting effects of acute activation of Hcrt neurons (*Carter et al., 2012*). However, the contribution

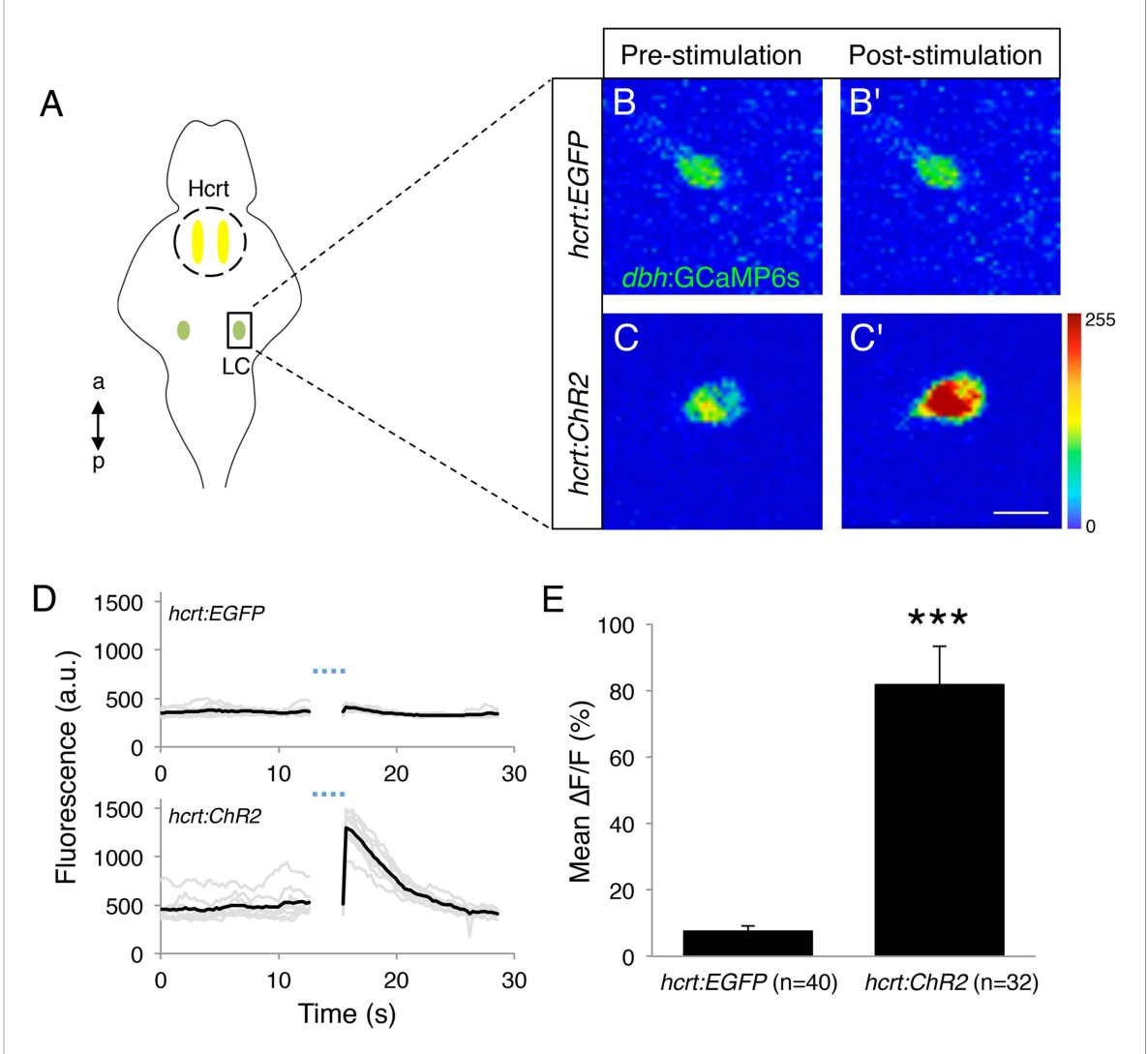

**Figure 7**. Optogenetic stimulation of Hcrt neurons activates LC neurons. (**A**) Schematic representation of areas stimulated (Hcrt, yellow) and imaged (LC, green). a = anterior, p = posterior. (**B**, **C**) Representative images of a LC cell expressing GCaMP6s in a *Tg(hcrt:EGFP)* (**B**) or *Tg(hcrt:ChR2-EYFP)* (**C**) larva before (**B**, **C**) and immediately after (**B'**, **C'**) stimulation of Hcrt neuron region. (**D**) GCaMP6s fluorescence intensity for a representative LC neuron in a *Tg(hcrt:EGFP)* (top) and a *Tg(hcrt:ChR2-EYFP)* (bottom) larva. Single-trial (gray) and average (black) responses are shown. (**E**) Mean ± s.e.m. ΔF/F (%) values averaged for all trials. n indicates number of trials for 5 *Tg(hcrt:EGFP)* and 4 *Tg(hcrt:ChR2-EYFP)* larvae. ***, p < 0.001 by one-way ANOVA. See **Videos 1 and 2** for examples of GCaMP6s imaging.

of NE to this phenotype is unclear. To test the hypothesis that NE is required for the arousing effect of Hcrt neuron stimulation, we assayed the behavioral effects of optogenetically activated Hcrt neurons in *dbh−/−* larvae (*Figure 8*). As we previously observed (*Figure 2D,E*), *dbh−/−* larvae were significantly less active during the 30 min of baseline recording than sibling controls (*dbh+/+* 31.0 ± 4.0, *dbh+/−* 39.7 ± 4.1 and *dbh−/−* 15.7 ± 2.6 s of activity, p < 0.05 by one-way ANOVA). *dbh−/−* larvae were also 25% less responsive to blue light than both *dbh+/+* and *dbh+/−* larvae (*Figure 8A,B*). However, when Hcrt neurons were activated optogenetically the requirement for NE became more pronounced, as the ChR2-induced increase in locomotor activity was reduced by 76% in *Tg(hcrt:ChR2-EYFP);dbh−/−* larvae compared to both *Tg(hcrt:ChR2-EYFP);dbh+/+* and *Tg(hcrt:ChR2-EYFP);dbh+/−* siblings (*Figure 8C,D*). Thus, while it is possible that supranormal optogenetic activation of Hcrt neurons could make NE more important for Hcrt-induced locomotor activity than it is under normal conditions, these results suggest that NE is an important effector of Hcrt-neuron induced arousal.

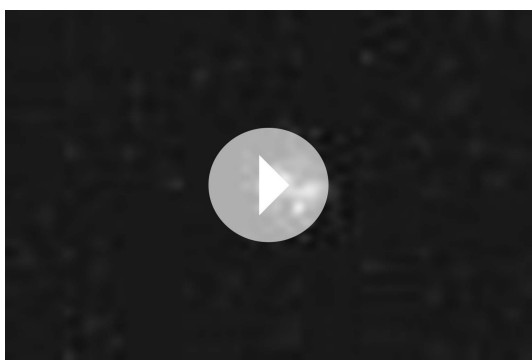

**Video 1.** Pulsed illumination of *hcrt:EGFP* neurons does not affect GCaMP6s fluorescence in LC neurons. Two trials for a representative LC neuron are shown. Frames labeled 'Stimulation' indicate periods during which the soma of Hcrt neurons were illuminated by ten 0.3 s pulses of 488 nm light, during which time GCaMP6s was not imaged in the LC. See *Figure 7* for quantification.

## Discussion

The first catecholamine to be identified as a neurotransmitter by Ulf von Euler in 1946, NE has been implicated in many aspects of physiology and behavior (*Berridge and Waterhouse, 2003*; *Weinshenker and Schroeder, 2007*; *Sara, 2009*). Several lines of evidence indicate that exogenous NE is a potent arousal-promoting agent (reviewed in *Berridge et al., 2012*). In *Drosophila*, lack of endogenous octopamine, which is considered the invertebrate equivalent of NE, results in increased sleep (*Crocker and Sehgal, 2008*). However, the role of endogenous NE in regulating vertebrate sleep remains unclear.

The murine *dbh* knockout (*Thomas et al., 1995*) displays no overt developmental defects. However, only 10% of *dbh*−/− pups produced by *dbh*+/− mothers survive embryonic development and no *dbh*−/− pups are born to *dbh*−/− mothers; all such embryos die by embryonic day 13.5 (*Thomas et al., 1995*). Of the *dbh*−/− pups that survive embryogenesis, only 40% reach adulthood. Although the basis of the embryonic lethality is unclear, it was hypothesized to be due to abnormal heart development or function, as the hearts of mutant embryos display greater heterogeneity in cell size and orientation (*Thomas et al., 1995*). Supplementation of drinking water during gestation with dihydroxyphenylserine (DOPS), which is converted to NE by L-aromatic-amino-acid decarboxylase (AADC), rescues the embryonic lethal phenotype. Following birth, DOPS supplementation is not required for continued survival of *dbh*−/− pups. Surprisingly, *dbh*−/− mice that reach adulthood were initially reported as having normal sleep/wake cycles (*Hunsley and Palmiter, 2003*), although a later study reported a 20% increase in sleep (*Ouyang et al., 2004*). These contradictory results could arise from differences in methodology, a relatively subtle and thus poorly reproducible sleep phenotype, or the complication that mice lacking *dbh* also exhibit higher levels of dopamine, the substrate of DBH and a major neurotransmitter in the brain.

In this study we sought to clarify the role of endogenous NE in regulating vertebrate sleep using pharmacology and genetics in zebrafish. Sleep can be distinguished from inactivity using electrophysiology or three behavioral criteria (*Campbell and Tobler, 1984*; *Borbely and Tobler, 1996*; *Allada and Siegel, 2008*). First, sleep primarily occurs during specific periods of the circadian cycle. Second, animals exhibit an increased arousal threshold during sleep, although they can still be aroused by strong stimuli, thus distinguishing sleep from paralysis or coma. Third, sleep is controlled by a homeostatic system, which can be demonstrated as an increased need for sleep following sleep deprivation. Based on these criteria, it has been shown that rest in a variety of organisms, including zebrafish (*Zhdanova et al., 2001*; *Prober et al., 2006*; *Yokogawa et al., 2007*), meets the behavioral definition of sleep. Several groups

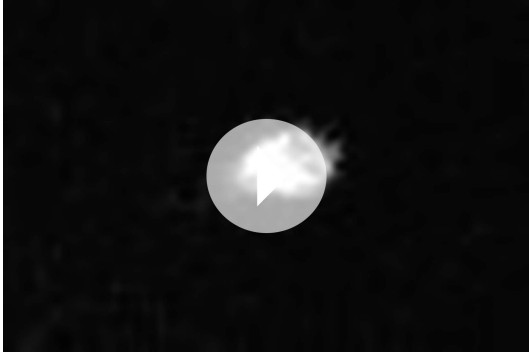

**Video 2.** Pulsed illumination of *hcrt:ChR2-EYFP* neurons increases GCaMP6s fluorescence in LC neurons. Two trials for a representative LC neuron are shown. Frames labeled 'Stimulation' indicate periods during which the soma of Hcrt neurons were illuminated by ten 0.3 s pulses of 488 nm light, during which time GCaMP6s was not imaged in the LC. See *Figure 7* for quantification.

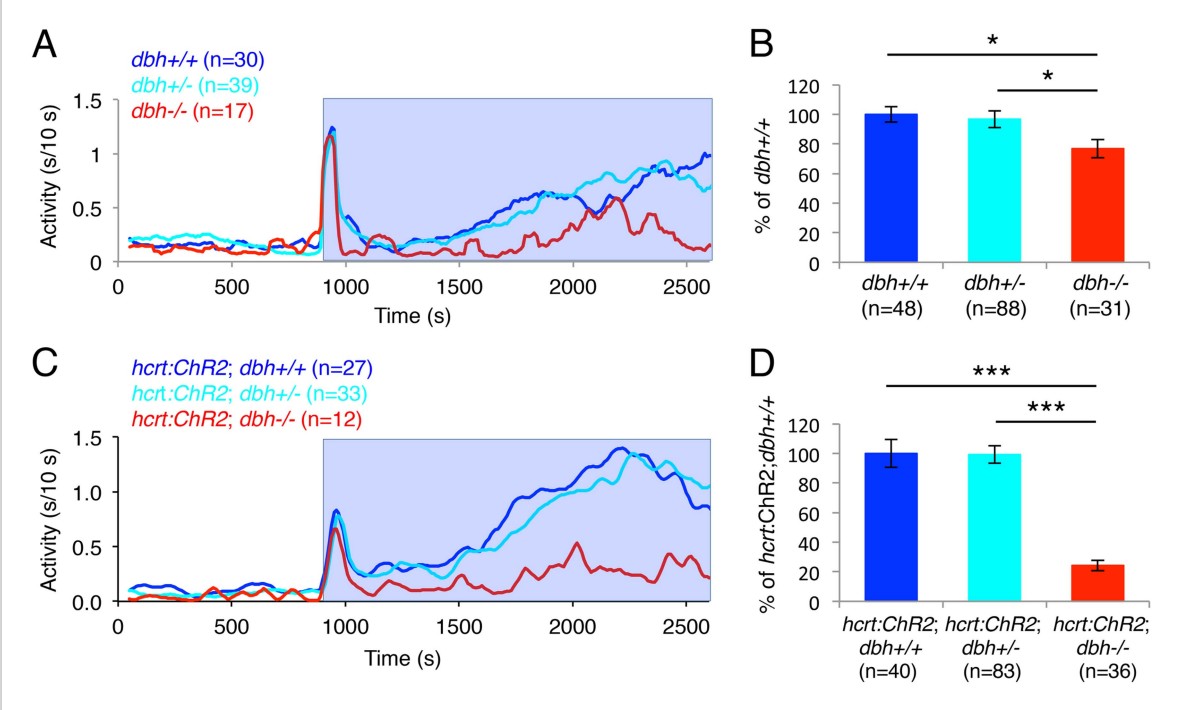

**Figure 8.** *dbh* is required for Hcrt neuron-induced increased locomotor activity. (**A**, **C**) Representative locomotor activity traces for *dbh–/–* and sibling control larvae without (**A**) and with (**C**) the *hcrt:ChR2-EYFP* transgene during blue light exposure. (**B**, **D**) Average locomotor activity relative to sibling controls. Data is pooled from 2 (**B**) or 3 (**D**) experiments. Data is represented as mean ± s.e.m. and is plotted as percentage of *dbh+/+* (**B**) or *hcrt:ChR2-EYFP; dbh+/+* (**D**). n indicates number of larvae. *, p < 0.05 and ***, p < 0.0001 by one-way ANOVA followed by Tukey's test to correct for multiple comparisons.

have demonstrated behavioral, anatomical, genetic and pharmacological conservation of sleep between zebrafish and mammals, establishing zebrafish as a simple vertebrate model for sleep research (*Zhdanova et al., 2001*; *Kaslin et al., 2004*; *Faraco et al., 2006*; *Prober et al., 2006*; *Renier et al., 2007*; *Yokogawa et al., 2007*; *Rihel et al., 2010*; *Gandhi et al., 2015*).

To study the function of NE in regulating zebrafish sleep, we generated a predicted null mutation in the single zebrafish *dbh* ortholog (*Figure 2—figure supplement 1*). Unlike mice, *dbh–/–* zebrafish larvae develop normally yet show reduced activity and strikingly increased sleep (+185% during the day and +57% during the night) compared to sibling controls (*Figure 2*). Importantly, these phenotypes were also observed in larvae treated with the alpha-adrenergic inhibitor prazosin (*Figure 1*), which blocks NE signaling without affecting dopamine levels. This observation suggests that the zebrafish *dbh–/–* phenotypes are due to lack of NE and not altered dopamine levels.

It is important to note that our pharmacological manipulations (blocking of the activating alpha1-adrenergic receptors with the antagonist prazosin; activation of the inhibitory alpha2-adrenergic receptors with the agonist clonidine; blocking of the activating alpha2-adrenergic receptors with the antagonist bopindolol) did not fully recapitulate the activity and sleep phenotypes observed in the *dbh* mutant. This is not surprising considering the inherent limitations of global exposure to a drug and the fact that each receptor type has at least five paralogs in zebrafish that may have different sensitivities to these drugs. Among these drugs, prazosin most faithfully phenocopied the *dbh* sleep/wake phenotype (reduced day and night activity and increased day and night sleep) with differences observed in sleep bout number. Clonidine reduced activity and increased sleep during the day, but had almost no effect at night. Bopindolol increased sleep during the day and night, but only inhibited activity at night. These observations suggest that alpha1-adrenergic receptors could be the main facilitators of noradrenergic modulation of larval zebrafish sleep.

Why do *dbh* mutant mice fail to exhibit the robust sleep phenotypes observed in NE deficient zebrafish? One possible explanation is that the chronic loss of NE in *dbh* mutant mice may induce

compensatory mechanisms during development that do not occur in response to chronic loss of NE in zebrafish. Another possibility is that DOPS treatment during gestation affects developmental processes that ultimately affect behavior. An intriguing third potential explanation involves the role of NE in the mammalian sympathetic nervous system in promoting brown adipose tissue (BAT) thermogenesis (*Cannon and Nedergaard, 2004*). BAT thermogenesis has been shown to promote slow wave sleep (*Dewasmes et al., 2003*) and to be required for sleep rebound after sleep deprivation (*Szentirmai and Kapas, 2014*). Therefore, genetic ablation of *dbh* could induce two competing effects. In the periphery, lack of NE reduces BAT thermogenesis and thus inhibits sleep. In the central nervous system, lack of NE interferes with the normal arousal function of the LC and thus inhibits wakefulness. The combination of these two opposing forces could result in a minor sleep increase in *dbh–/–* mice, and the magnitude of this phenotype might be particularly sensitive to experimental conditions. However, zebrafish larvae and flies do not regulate their body temperature and are thus subject only to the sleep promoting effects of NE ablation, resulting in a dramatic increase in sleep. It would be interesting to test whether mice in which *dbh* is deleted only centrally, and thus have normal BAT thermogenesis in the periphery, display a similarly dramatic sleep increase.

To further characterize the *dbh* phenotype, we used an automated arousal threshold assay using a mechano-acoustic stimulus. We were surprised to find that although *dbh–/–* larvae sleep more, they display a lower arousal threshold (i.e. they respond to stimuli of lower intensity), as well as a higher maximal response than sibling controls (*Figure 3A*). Genetic ablation of *dbh* is predicted to increase dopamine levels in neurons that normally express *dbh,* since this enzyme converts dopamine to NE. Indeed, dopamine levels are higher in *dbh* mutant mice compared to sibling controls (*Thomas et al., 1995*). However, pharmacological inhibition of adrenergic receptors should not interfere with the conversion of dopamine to NE. Interestingly, we were able to pharmacologically replicate the reduced arousal phenotype seen in *dbh–/–* animals by treatment with bopindolol, a beta-adrenergic inhibitor, but not with the alpha1-adrenergic inhibitor prazosin, or the alpha2-adrenergic agonist clonidine. The ability of bopindolol to recapitulate the reduction in arousal threshold seen in the *dbh* mutant suggests that this aspect of the mutant phenotype is a consequence of the silencing of beta-adrenergic receptor signaling. Furthermore, the inability of prazosin (which increases sleep during day and night) or clonidine (which increases sleep during the day) to recapitulate the phenotype suggests that the observed reduction in arousal threshold is not simply a consequence of lighter sleep due to 'over-sleeping' in the *dbh* mutant. These observations suggest that lack of beta-adrenergic receptor signaling potentiates responses to sensory stimuli, and are particularly interesting considering that NE reuptake inhibitors such as atomoxetine, which increase NE signaling, are used to treat patients suffering from Attention Deficit/Hyperactivity Disorder (ADHD) (*Garnock-Jones and Keating, 2009*). It is important to note that these pharmacological results do not preclude a role of excess dopamine release in the *dbh* mutant by the previously noradrenergic neurons of the LC and/or medulla oblongata in reducing the arousal threshold. This hypothesis is supported by zebrafish studies implicating dopamine in arousal modulation (*Burgess and Granato, 2007*; *Mu et al., 2012*) as well as work demonstrating that dopaminergic stimulation during anesthesia produces a robust arousal response (*Taylor et al., 2013*). Furthermore, dopamine agonists have been shown to increase locomotion in zebrafish larvae (*Rihel et al., 2010*; *Irons et al., 2013*) and dopamine promotes locomotor development in zebrafish larvae (*Lambert et al., 2012*).

The Hcrt neurons of the hypothalamus are a major arousal promoting center. Several lines of evidence suggest that Hcrt promotes arousal, at least in part, via the LC. First, Hcrt neurons send dense projections to the LC in rodents and zebrafish (*Peyron et al., 1998*; *Chemelli et al., 1999*; *Date et al., 1999*; *Horvath et al., 1999*; *Kaslin et al., 2004*; *Prober et al., 2006*) (*Figure 4—figure supplement 1A,B*). Second, the LC expresses the Hcrt receptor in rodents (*Horvath et al., 1999*; *Bourgin et al., 2000*) and zebrafish (*Prober et al., 2006*). Third, application of Hcrt peptide depolarizes LC neurons in brain slices (*Hagan et al., 1999*) and in vivo (*Bourgin et al., 2000*). Fourth, optogenetic activation of Hcrt neurons induces c-Fos expression in the LC of mice (*Carter et al., 2012*) and GCaMP6 activation in the LC of zebrafish larvae (*Figure 7*). Fifth, acute inhibition of LC neurons blocks the effect of acute Hcrt neuron activation on sleep to wake transitions (*Carter et al., 2012*). While these studies suggest that the LC plays an important role in mediating Hcrt-induced arousal, it was unknown whether NE, a neurotransmitter in the LC, is required for this process. This question is important because the LC produces other neurotransmitters and neuropeptides that have been shown to affect sleep, including dopamine (*Dzirasa et al., 2006*), neuropeptide Y (*Dyzma et al., 2010*), neurotensin (*Erwin and Radcliffe, 1993*) and vasopressin (*Born et al., 1992*).

To address this question, we used two independent strategies to drive Hcrt-induced arousal. First, we overexpressed Hcrt using a heat-shock inducible promoter, which was previously shown to decrease sleep in zebrafish larvae (*Prober et al., 2006*). We observed that in the absence of NE, sleep reduction was attenuated (*Figure 4*). Second, we developed a large-scale and non-invasive optogenetic assay and used it to activate Hcrt neurons in freely behaving larvae (*Figure 5*). Similar to Hcrt overexpression, activation of Hcrt neurons increased locomotor activity, consistent with the increased sleep-to-wake transitions previously described in mammals (*Adamantidis et al., 2007*). This observation provides evidence of a causal relationship between the activity of Hcrt neurons and locomotor activity in zebrafish, and together with data from the mouse (*Adamantidis et al., 2007*) supports an evolutionarily conserved role of Hcrt neurons in promoting arousal. We found that the arousing effects of Hcrt overexpression (*Figure 4*) and Hcrt neuron activation (*Figure 8*) were dramatically reduced in *dbh* mutant larvae, indicating that NE is an important downstream effector of Hcrt-mediated arousal.

Interestingly, lack of NE did not completely abolish arousal induced by Hcrt overexpression (*Figure 4*), suggesting that Hcrt also promotes arousal via NE-independent pathways. Whether these pathways employ other molecules produced by the LC or are LC-independent remains to be tested. Zebrafish Hcrt neurons also send projections to noradrenergic cells in the medulla oblongata, similar to mammals (*Ciriello et al., 2003*; *Zhang et al., 2004*), suggesting that these cells might also mediate Hcrt-induced arousal. Furthermore, in humans, Hcrt neurons are reciprocally connected to other important arousal centers such as the cholinergic basal forebrain, the histaminergic tuberomammillary nucleus, the dopaminergic ventral tegmental area and the serotonergic dorsal raphe (*Alexandre et al., 2013*). Many of these connections are also present in zebrafish, with Hcrt neurons projecting to dopaminergic, histaminergic and serotonergic populations (*Panula et al., 2010*). Although functional interactions between Hcrt neurons and these populations have yet to be explored in zebrafish, the anatomical and molecular similarities of zebrafish and mammalian brains suggest that at least a portion of Hcrt-induced arousal is likely to be mediated by these centers.

In summary, we have demonstrated that endogenous NE is required to maintain normal levels of wakefulness in a diurnal vertebrate. This study, together with previous reports in *Drosophila* (*Crocker and Sehgal, 2008*) and mice (*Ouyang et al., 2004*), support a phylogenetically conserved role for endogenous NE in promoting wakefulness. Furthermore, we showed that NE also plays an important role in mediating arousal induced by either overexpression of the Hcrt peptide or activation of Hcrt neurons. Finally, we established and characterized the zebrafish *dbh* mutant, a useful tool for studying other processes that are thought to involve NE, including the fight-or-flight response (*Colwill and Creton, 2011*), congestive heart failure (*Thomas and Marks, 1978*), cognitive disorders such as Alzheimer's (*Chalermpalanupap et al., 2013*) and Parkinson's diseases (*Rommelfanger et al., 2007*), and neuropsychiatric disorders including depression and ADHD (*Chamberlain and Robbins, 2013*).

## Materials and methods

### Ethics statement
All experiments followed standard protocols (*Westerfield, 2000*) in accordance with the California Institute of Technology Institutional Animal Care and Use Committee guidelines (animal protocol 1580).

### Transgenic and mutant zebrafish
#### Tg(hcrt:ChR2(H134R)-EYFP)
A 1 kilobase (kb) genomic fragment upstream of zebrafish *hcrt*, described previously (*Faraco et al., 2006*), was amplified using the primers 5′-ATAATAAATAAATCTGATGGGGTTTT-3′ and 5′-GAGTT-TAGCTTCTGTCCCCTG-3′, and subcloned 5′ to a transgene encoding the H134R variant of channelrhodopsin-2 (ChR2) fused to EYFP (*Nagel et al., 2005*; *Lin et al., 2009*), in a plasmid containing Tol2 transposase recognition sequences. This plasmid was co-injected with Tol2 *transposase* mRNA to generate *Tg(hcrt:ChR2-EYFP)* stable transgenic lines. Screening for EYFP expression identified several stable lines, two of which were used for experiments. Both lines expressed EYFP specifically in Hcrt neurons. One line had a single insertion of the transgene and exhibited weak EYFP fluorescence. For the experiment described in *Figure 5*, animals heterozygous

for the transgene were incrossed to compare sibling larvae exhibiting no EYFP (wild-type) and moderate EYFP (homozygous transgenic) fluorescence. Stimulation of Hcrt neurons using ChR2 had a much stronger effect on locomotor activity in homozygous transgenic animals compared to heterozygotes. We did not use this line in combination with *hcrtr* and *dbh* mutants in the same experiment because the number of larvae of each genotype from a mating was too low to obtain robust data. The second line contained multiple insertions of the transgene and outcrossing these fish produced larvae with EYFP fluorescence intensities ranging from weak to strong. We used this line for experiments that required outcrossing to *hcrtr* or *dbh* mutants in order to obtain larvae with strong ChR2 expression. Adults from this line were mated with *hcrtr or dbh* mutants, and larvae with strong EYFP fluorescence were raised to adulthood. These heterozygous mutants were then incrossed to generate larvae for behavioral experiments. All larvae expressed ChR2-EYFP with varying fluorescent intensities due to multiple insertions of the transgene. At the end of each behavioral experiment, EYFP fluorescence was visualized and only larvae with bright fluorescence were used for analysis. To determine the effect of blue light stimulation on *hcrtr* or *dbh* mutants without the ChR2-EYFP transgene, heterozygous mutant fish were incrossed to obtain homozygous mutant, heterozygous mutant and wild type sibling larvae.

For Hcrt overexpression experiments, fish heterozygous for the *hsp:Hcrt* transgene and for the *hcrtr* or *dbh* mutation were mated to fish that were homozygous for the *hcrtr* or *dbh* mutation. The behavior of larvae with or without the *hsp:Hcrt* transgene was then compared in heterozygous and homozygous mutants. This mating scheme was used to ensure that each experiment had enough larvae of each genotype to obtain robust data. The *hcrtr* mutant (*Yokogawa et al., 2007*), *dbh* mutant (*Chen et al., 2013a*), Tg(dbh:EGFP) transgenic line (*Liu et al., 2015*), Tg(hcrt:RFP) transgenic line (*Liu et al., 2015*) and Tg(hsp:Hcrt) transgenic line (*Prober et al., 2006*) have previously been described. Tg(hsp:Hcrt) fish were genotyped using the primers 5′-CGGGACCACCATGGACT-3′ and 5′-GGTTTGTCCAAACTCATCAATGT-3′, which generate a 470 bp band. *dbh* mutant fish were genotyped using the primers 5′-GAGCTCATGCAACGAAC-3′, 5′-GTAAAACTCAACTGTTTACC TAAAG-3′ and 5′-GTGCGTACATCTTTCGGG -3′, which generate 1 band for wild type (198 bp), two bands for homozygous mutant (113 bp and 202 bp) and three bands for heterozygous mutant (113 bp, 198 bp and 202 bp). *hcrtr* mutant fish were genotyped using the primers 5′-CCACC CGCTAAAATTCAAAAGCACTGCTAAC-3′ and 5′-CATCACAGACGGTGAACAGG-3′, which generate a 170 bp product that is cut by DdeI (New England Biolabs, Ipswich, MA, United States) in the mutant to produce 140 and 30 bp bands. We outcrossed the *dbh* mutant to the parental TLAB wild type strain three times to reduce the possibility of unlinked mutations, and we did not observe developmental defects that can be indicative of such mutations. During the course of routine genotyping by fin clipping, we found that adult *dbh−/−* fish often die following anesthesia with tricaine (E10521, Sigma-Aldrich, St. Louis, MO, United States) at a commonly used dose (0.16 mg/ml). A lower concentration of tricaine (0.08 mg/ml) allowed survival of *dbh−/−* fish. Furthermore, *dbh−/−* fish sometimes died during or shortly after mating, perhaps due to a heightened sensitivity to stress. *dbh−/−* larvae respond normally to sensory stimuli such as gentle poking.

## Sleep/wake analysis

Sleep/wake analysis was performed as previously described (*Prober et al., 2006*). Larvae were raised on a 14/10 hr light/dark (LD) cycle at 28.5°C with lights on at 9 am and off at 11 pm. Dim white light was used to raise larvae for optogenetic experiments to prevent stimulation of ChR2 by ambient light. Individual larvae were placed into each well of a 96-well plate (7701-1651, Whatman, Pittsburgh, PA, United States) containing 650 μl of E3 embryo medium (5 mM NaCl, 0.17 mM KCl, 0.33 mM $CaCl_2$, 0.33 mM $MgSO_4$, pH 7.4). Locomotor activity was monitored using a videotracking system (Viewpoint Life Sciences, Lyon, France) with a Dinion one-third inch Monochrome camera (Dragonfly 2, Point Grey, Richmond, Canada) fitted with a variable-focus megapixel lens (M5018-MP, Computar, Cary, NC, United States) and infrared filter. The movement of each larva was recorded using the quantization mode. The 96-well plate and camera were housed inside a custom-modified Zebrabox (Viewpoint Life Sciences) that was continuously illuminated with infrared lights. The 96-well plate was housed in a chamber filled with recirculating water to maintain a constant temperature of 28.5°C. The parameters used for movement detection were: detection threshold, 15; burst, 29; freeze, 3; bin size, 60 s. Data were analyzed using custom Perl and Matlab (Mathworks, Natick, MA, United States) scripts (*Source code 1*), which conform to the open source definition.

For Hcrt overexpression experiments, videotracker analysis was initiated at 4 dpf. During the afternoon of 6 dpf, the 96-well plate was transferred to a 37°C water bath for 1 hr to induce Hcrt overexpression. For each larva, total sleep during night 6 was divided by the average total sleep on night 5 for all larvae of the same genotype and converted to a percentage to compare among different genotypes.

## Pharmacology

Prazosin hydrochloride (P7791, Sigma–Aldrich), clonidine hydrochloride (C7897, Sigma–Aldrich) and bopindolol malonate (SC-200144, Santa Cruz Biotechnology, Dallas, TX, United States) were dissolved in dimethyl sulfoxide (DMSO, 4948-02, Macron Chemicals, Center Valley, PA, United States) and added to E3 medium for a final concentration of 0.1% DMSO and 100 µM prazosin, 5 µM clonidine or 20 µM bopindolol. At these concentrations, we observed robust behavioral phenotypes without apparent toxicity or abnormal responses to a gentle stimulus. Controls were exposed to 0.1% DMSO alone. For sleep/wake experiments drugs were added during the evening of the fourth day of development and recording was performed from the beginning of day 5 until the end of night 6. For arousal experiments, drugs were added in the afternoon of day 5, and experiments were performed during night 5 (12:30 am to 7:30 am).

## Analysis of *dbh* expression and NE levels

*dbh* ISH was performed using standard protocols (*Thisse and Thisse, 2008*) and developed using nitro-blue tetrazolium and 5-bromo-4-chloro-3′-indolyphosphate (10760978103, Roche, Mannheim, Germany). A fragment of the *dbh* gene was used as a probe (*Guo et al., 1999*). NE levels were measured using an ELISA assay (17-NORHU-E01-RES, ALPCO, Salem, NH, United States) according to the manufacturer's instructions. Five larvae were analyzed per sample, with four samples analyzed in triplicate for each genotype.

## Arousal threshold assay

The videotracking system was modified by adding an Arduino (http://www.arduino.cc/) based automated driver to control two solenoids (28P-I-12, Guardian Electric, Woodstock, IL, United States) that delivered a tap to a 96-well plate containing larvae. This setup allowed us to drive the solenoids with voltage ranging from 0 V to 20 V over a range of 4095 settings (from 0.01 to 40.95). In our experiments we used taps ranging from a power setting of 1–36.31. Taps of 14 different intensities were applied in a random order from 12:30 am to 7:30 am during the fifth night of development with an inter-trial-interval of 1 min. Previous studies showed that a 15 s interval between repetitive stimuli is sufficient to prevent behavioral habituation (*Burgess and Granato, 2007*; *Woods et al., 2014*). The background probability of movement was calculated by identifying for each genotype the fraction of larvae that moved 5 s prior to all stimuli delivered during an experiment (14 different tap powers x 30 trials per experiment = 420 data points per larva; average background movement). This value was subtracted from the average response fraction value for each tap event (corrected response = average response – average background movement). The response of larvae to the stimuli was monitored using the videotracking software and was analyzed using Matlab (Mathworks) and Excel (Microsoft, Redmond, WA, United States). Statistical analysis was performed using the Variable Slope log(dose) response curve fitting module of Prism (Graphpad, La Jolla, CA, United States).

## Immunofluorescence

Samples were fixed in 4% paraformaldehyde (PFA) in PBS overnight at 4°C and then washed with 0.25% Triton X-100/PBS (PBTx). Brains were manually dissected and blocked for at least 1 hr in 2% goat serum/2% DMSO/PBTx at room temperature or overnight at 4°C. Antibody incubations were performed in blocking solution overnight at 4°C. Primary antibodies were rabbit anti-orexin A (AB3704, 1:500; Millipore, Temecula, CA, United States), rabbit anti-RFP (632496, 1:100, Clontech, Mountain View, CA, United States) and chicken anti-GFP (GFP-1020, 1:400, AvesLabs, Tigard, OR, United States). Alexa Fluor secondary antibodies were used (1:500 for anti-orexin and anti-RFP and 1:600 for anti-GFP, Life Technologies, Carlsbad, CA, United States). Samples were mounted in 50% glycerol/PBS and imaged using a Zeiss LSM 780 confocal microscope (Zeiss, Oberkochen, Germany).

## ISH

To exclude the possibility that the HS promoter response is suppressed in *hcrtr* and *dbh* mutants, we performed ISH to compare the level of overexpressed *hcrt* mRNA in *hcrtr* and *dbh* homozygous mutants to their heterozygous mutant siblings. Fish heterozygous for the *hsp:Hcrt* transgene and for the *hcrtr* or *dbh* mutation were mated to *hcrtr* or *dbh* homozygous mutants, respectively. Larvae from these crosses were heat shocked during the afternoon at 6 dpf to induce *hcrt* overexpression, and were fixed 30 min after HS in 4% PFA overnight at room temperature. ISH was performed using digoxygenin (DIG) labeled antisense riboprobes (*Thisse and Thisse, 2008*). Images were acquired using a Zeiss AxioImagerM1 microscope and samples were then genotyped by PCR.

## Assay for activation of *hcrt*-expressing neurons by ChR2

*Tg(hcrt:ChR2-EYFP)* larvae were placed in a 96 well plate in the videotracking system as described above, left in the dark for 8 hr, and then exposed to blue light for 30 min starting at 1 am. Larvae were then fixed in 4% PFA overnight at 4°C. Double-fluorescent ISH was performed using DIG- and 2,4-dinitrophenol (DNP)-labeled riboprobes and the TSA Plus DNP System (NEL747A001 KT, PerkinElmer, Waltham, MA, United States). Probes specific for *c-fos* and *eyfp* were used to quantify the number of *eyfp*-expressing Hcrt neurons that expressed *c-fos*. *Tg(hcrt:EGFP)* larvae (*Prober et al., 2006*) were used as negative controls. Samples were mounted in 50% glycerol/PBS and imaged using a Zeiss 780 LSM confocal microscope.

## Optogenetic behavioral assay and analysis

The videotracking system was modified to include a custom array containing three sets of red and blue LEDs (627 nm, MR-D0040-10S and 470 nm, MR-B0040-10S, respectively, Luxeon V-star, Brantford, Canada) mounted 15 cm above and 7 cm away from the center of the 96-well plate to ensure uniform illumination. The LEDs were controlled using a custom built driver and software written in BASIC stamp editor. A power meter (1098293, Laser-check, Santa Clara, CA, United States) was used before each experiment to verify uniform light intensity (~400 μW at the surface of the 96-well plate) and similar red and blue light intensities were used. During the afternoon of the fifth day of development, single larvae were placed into each well of a 96-well plate as described above and placed in the videotracker in the dark for 8 hr. Larvae were then exposed to either red or blue light for 30 min, starting at 1 am. Three trials were performed during the night, with an inter-trial interval of 3 hr. Total activity for each larva was monitored for 30 min before and after light onset, with data collected in 10 s bins. Light onset caused a short burst of locomotor activity lasting for ~30 s for all genotypes, so data obtained during the minute before and after light onset was excluded from analysis. A large burst of locomotor activity was also observed for all genotypes when the lights were turned off after the 30 min illumination period. This data was excluded from analysis and is not shown in the figures. The total amount of locomotor activity of each larva during the 30 min of light exposure, excluding the minute after light onset, was divided by the average baseline locomotor activity for all larvae of the same genotype. The baseline period was defined as 30 min before light onset, excluding the minute before light onset. Data from multiple experiments were pooled and converted to percentage of wild type larvae.

## GCaMP6s imaging

A 1.1 kb genomic fragment upstream of the *dbh* gene was amplified using the primers 5′-ACTTGAACCAGCGACCTTCT-3′ and 5′-GGTTTGAAGGCCTTTCTAAGTTTTT-3′ (*Liu et al., 2015*) and cloned 5′ to a transgene encoding GCaMP6s (*Chen et al., 2013b*) in a plasmid containing Tol2 transposase recognition sequences. Either *Tg(hcrt:ChR2-EYFP)* or *Tg(hcrt:EGFP)* embryos were injected at the 1–2 cell stage with a solution containing 50 ng/μl plasmid, 0.04% phenol red and 50 ng/μl Tol2 *transposase* mRNA. This injection procedure resulted in GCaMP6s expression in no more than 1 LC neuron per animal. At 5 dpf, larvae were paralyzed by immersion in 1 mg/ml α-bungarotoxin (2133, Tocris, Bristol, UK) dissolved in E3, embedded in 1% low melting agarose (EC-202, National Diagnostics, Atlanta, GA, United States) and imaged using a 20× water immersion objective on a Zeiss LSM 780 confocal microscope. To stimulate Hcrt neurons, a region of interest (ROI) that encompassed the Hcrt neuron soma was illuminated using a 488 nm laser at 100% power. Ten pulses lasting 0.3 s each were applied over 3.2 s using the bleaching function, and then a ROI encompassing a GCaMP6s-expressing LC soma was imaged at 4 Hz for 30 s. The time between the final stimulation pulse and

initiation of imaging was less than 0.1 s. This cycle of stimulation and imaging was repeated 8 times for each GCaMP6s-expressing soma. A baseline of 60 s was recorded before the first stimulation. Movies were processed using ImageJ (*Schneider et al., 2012*). The mean fluorescence of each LC neuron was measured by drawing a ROI around the soma. Baseline fluorescence ($F_o$) for each trial was defined as 10 frames immediately preceding the stimulation, and 10 frames post stimulation (F) were used to measure the total change in fluorescence ($\Delta F/F_o = (F-F_o)/F_o$).

## Acknowledgements

We thank Mike Walsh for assistance with the optogenetic and tapping assays, Andres Collazo for imaging assistance, Viveca Sapin and Jae Engle for genotyping assistance, David Schoppik, Alix Lacoste and Owen Randlett for tapping assay advice and Catherine Oikonomou for critical reading of the manuscript. This work was supported by grants from the NIH (GO: F32NS082010; DAP: NS060996, NS070911 and DA031367), the Mallinckrodt Foundation, the Rita Allen Foundation and the Brain and Behavior Research Foundation (DAP). We declare no competing interests.

## Additional information

### Funding

| Funder | Grant reference | Author |
|---|---|---|
| Rita Allen Foundation | | David A Prober |
| Mallinckrodt Foundation | | David A Prober |
| Brain and Behavior Research Foundation | | David A Prober |
| National Institutes of Health (NIH) | F32 NS082010 | Grigorios Oikonomou |
| National Institutes of Health (NIH) | R00 NS060996 | David A Prober |
| National Institutes of Health (NIH) | R01 NS070911 | David A Prober |
| National Institutes of Health (NIH) | R01 DA031367 | David A Prober |

The funders had no role in study design, data collection and interpretation, or the decision to submit the work for publication.

### Author contributions

CS, GO, Conception and design, Acquisition of data, Analysis and interpretation of data, Drafting or revising the article; DAP, Conception and design, Drafting or revising the article

### Author ORCIDs

David A Prober, http://orcid.org/0000-0002-7371-4675

### Ethics

Animal experimentation: All experiments followed standard protocols (*Westerfield, 2000*) in accordance with the California Institute of Technology Institutional Animal Care and Use Committee guidelines.

## Additional files

### Supplementary file

• Source code 1. Scripts used for analysis of behavioral data. *sort_fish_sttime_192.pl* is a Perl script (*Prober et al., 2006*) that converts data acquired by the Viewpoint videotracker system to a format that is useful for analysis using Matlab and removes notations that are not relevant to behavioral analysis. *perl_batch_192well.m* is a Matlab script that allows the *sort_fish_sttime_192.pl* script to run on the Matlab platform. *TapAnalysis.m* is a Matlab script that analyzes tapping assay data and generates a table that lists the number of larvae that moved during each tapping event. *VT_analysis.m* is a Matlab script (modified from *Prober et al., 2006*) that analyzes locomotion data collected by the Viewpoint videotracker system to quantify several metrics, including activity, waking activity, sleep, sleep architecture and sleep latency. Detailed instructions on the use of these scripts will be provided upon request.

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
