## [Decision Letter]

Thank you for sending your work entitled “Norepinephrine is Required to Promote Wakefulness and for Hypocretin-Induced Arousal in Zebrafish” for consideration at *eLife*. Your article has been favorably evaluated by Eve Marder (Senior editor) and three reviewers, one of whom, Louis Ptáček, is a member of our Board of Reviewing Editors.

The Reviewing editor and the other reviewers discussed their comments before we reached this decision, and the Reviewing editor has assembled the following comments to help you prepare a revised submission.

The manuscript by Singh and colleagues is a nice piece of work that is logical in its progression. One very novel and powerful technical advance is the in vivo optogenetics in a 96 well format. It would be better to not refer to this as “high-throughput” as many think of HTP in the context of biochemical/fluorescent assays that allow easy assessment of many thousand or more of assays. This is clearly HTP for a behavioral analysis though, and the authors could simply tone down the term. This approach takes advantage of the transparency of fish and the ability to excite proteins without surgical implantation that is necessary in mice. This study follows logically from examination of HCRT->HCRTR->NE signaling on “wakefulness”. The bottom line is that hypocretin promotes arousal through its receptor in the locus coeruleus (LC) and specifically on NE-expressing cells in the LC. Since the investigators are not technically looking at sleep, more careful language (rest-activity rhythms, “sleep-like behavior”) would be more appropriate. Many of us believe rest activity as described here will be conserved (although not identical) to mammalian sleep. Of course, there will be some periods of inactivity when an animal is awake, thus, the need for greater care with the semantics.

Essential revisions:

1) Prazosin is an alpha1 specific antagonist. Its use leads to decreased activity in fish, increased number of bouts of sleep-like behavior, and decreased latency. However, there is no discussion of other receptors, namely alpha2, beta1, and beta2. The *dbh-/-* experiment looks at the effect of decreasing NE which would have effects on all receptors.

2) Related to #1, the authors don't discuss the discrepancy in Prazosin and *dbh* experiments in panels H of Figures 1 and 2. Prazosin has no effect on bout length at night while it is significantly increased in the *dbh* experiment. How many (if any) paralogous genes exist in zebrafish? Is there redundancy? It is possible that this may relate to alpha1-specific affects vs. global NE affects. One additional experiment is to use clonidine (alpha2 agonist) (or the IV form, Dexmedetomidine), or beta1 antagonists (beta-blockers) to help dissect differential effects on different receptors.

3) The result that adult *dbh-/-* fish have higher mortality related to normal doses of the anesthetic tricaine is interesting, yet out of place and unsupported by the data provided. You may consider omitting this finding, or explaining better its relevance.

4) With respect to the prazosin experiments, the investigators overstate the case for the effect not being dopamine-mediated. This would require measuring dopamine levels directly. It would be worthwhile to review (and to reference) the work from Ken Solt's lab showing that DA actions in VTA are important for arousal from anesthesia and to tone down the conclusion to “suggests the affect in not dopamine-mediated”, this may also be useful in the discussion surrounding Figure 3.

5) The paper claims in several places that “NE is required for Hcrt-induced arousal” (this quote is from the Abstract). This is not true. There is a major quantitative effect in the mutant (and see below) but as the arousal enhancement is not eliminated, this bold statement is incorrect.

6) It would have been good had they used the drug prazosin in combination with the Hcrt-induced arousal rather than just the mutant fish alone. NE elimination need not be the only effect of the KO, especially as it is systemic and all through development. For example, there could be compensatory changes in the endocrine system. Replication with the drug would increase confidence in the conclusion.

7) How long was the pretreatment in Figure 1 (how fast-acting is the drug), was a dose-response curve done (where is 100 μM on this curve?), and was the drug used in the *db*h KO fish? Was it without effect as one might predict if it is specific for alpha-adrenergic receptors?

8) The arousal threshold effects (Figure 3) are rather modest, and enigmatic. First in Figure 3, the “increased response rate” (not a response rate, just an enhanced response) is probably just an independent effect as the authors suggest. However, this means that they should replot their data subtracting a fixed value from each mutant response, and then compare with the WT. There may be a relatively small residual effect. Even in Figure 3, it is difficult to interpret the magnitude of the effect because it is not clear what the X-axis is (log base 2, base 10?). The fact that the drug and the mutant effects are similar (even if modest) appears to underscore the conclusion but still… And if the drug result (3B) is really more impressive than the mutant (1A), what does this do to the dopamine hypothesis? Perhaps the authors could consider that the excess sleep itself lowers the arousal threshold. (Too much sleep; sleep more lightly?). And the more sleep itself is no surprise. Is this possible?

9) The role of the LC is also no surprise given what has been done in rodents (e.g., [13]). It is however of value to pinpoint the role of NE within the LC. Given that the effect is far from complete, it means that there are other arousal systems that probably lie downstream of Hcrt – as indicated in the literature. It would be good if the authors discussed some of these and the likelihood that they also play a role in fish.

10) The authors spend considerable effort emphasizing the controversy in the sleep phenotype of mouse *dbh* knockouts. While it is true that two groups reported different sleep results for those mice, it is unclear whether this fish study truly resolves the controversy – it only provides supporting evidence that NA is wake-promoting in a phylogenetic sense and is consistent with analysis of the role of octopamine in *Drosophila* sleep. The evidence they provide for hypocretin promoting arousal partially via NA is a nice extension of a prior study done in mice by [13], which implicated LC neurons in the arousing effects of hypocretin.

---

## [Author Response]

*The manuscript by Singh and colleagues is a nice piece of work that is logical in its progression. One very novel and powerful technical advance is the* in vivo *optogenetics in a 96 well format. It would be better to not refer to this as “high-throughput” as many think of HTP in the context of biochemical/fluorescent assays that allow easy assessment of many thousand or more of assays. This is clearly HTP for a behavioral analysis though – and the authors could simply tone down the term. This approach takes advantage of the transparency of fish and the ability to excite proteins without surgical implantation that is necessary in mice. This study follows logically from examination of HCRT->HCRTR->NE signaling on “wakefulness”. The bottom line is that hypocretin promotes arousal through its receptor in the locus coeruleus (LC) and specifically on NE-expressing cells in the LC. Since the investigators are not technically looking at sleep, more careful language (rest-activity rhythms, “sleep-like behavior”) would be more appropriate. Many of us believe rest activity as described here will be conserved (although not identical) to mammalian sleep. Of course, there will be some periods of inactivity when an animal is awake, thus, the need for greater care with the semantics*.

We thank the reviewers for their favorable assessment of our study and their insightful suggestions that have helped us to significantly improve the manuscript. We have replaced the term “high-throughput” with “large-scale” because this term is consistent with our ability to assay 96 animals simultaneously, which is a much larger scale behavioral experiment compared to the 5-10 (or fewer) rodents that are typically studied, without evoking the context of high-throughput assays that involve thousands of samples. We also now describe the optogenetic assay as non-invasive, which contrasts with the invasive nature of rodent optogenetic assays, as the reviewers note. Please see the end of our response for a discussion of “sleep” terminology.

*1) Prazosin is an alpha1 specific antagonist. Its use leads to decreased activity in fish, increased number of bouts of sleep-like behavior, and decreased latency. However, there is no discussion of other receptors, namely alpha2, beta1, and beta2. The* dbh-/- *experiment looks at the effect of decreasing NE which would have effects on all receptors.*

Please see comment #2.

*2) Related to #1, the authors don't discuss the discrepancy in Prazosin and* dbh *experiments in panels H of*
Figures 1 and 2*. Prazosin has no effect on bout length at night while it is significantly increased in the* dbh *experiment. How many (if any) paralogous genes exist in zebrafish? Is there redundancy? It is possible that this may relate to alpha1-specific affects vs. global NE affects. One additional experiment is to use clonidine (alpha2 agonist) (or the IV form, Dexmedetomidine), or beta1 antagonists (beta-blockers) to help dissect differential effects on different receptors*.

Zebrafish have a total of five alpha1-adrenergic receptors (*adra1aa*, *adra1ab*, *adra1ba*, *adra1bb*, *adra1d*). alpha1-adrenergic receptors activate phospholipase C causing a rise of intracellular Ca^2+^ levels. It is unlikely that all five zebrafish receptors are inhibited by prazosin to the same extent. The differences we observe between the *dbh* mutant and the prazosin treated animals are likely a combination of both imperfect inactivation of all alpha1-adrenergic receptors by prazosin, as well as prazosin having no effect on the alpha2 and beta-adrenergic receptors.

Zebrafish have five alpha2-adrenergic receptors (*adra2a*, *adra2b*, *adra2c*, *adra2aa*, *adra2ab*) and six beta-adrenergic receptors (*adb1*, *adrbk2*, *adrb2a*, *adrb2b*, *adrb3a*, *adrb3b*). alpha2 receptors are inhibitory and their activation leads to reduction of cAMP and Ca^2+^ levels. Activation of beta-adrenergic receptors results in increased cAMP levels. As suggested by the reviewer, we have added new experiments in which we used the alpha2 agonist clonidine and the beta1 antagonist bopindolol to investigate the role of these receptors in sleep regulation. The main effect of clonidine treatment is a reduction in day activity and increase in day sleep, with no effect in night activity or sleep (Figure 1—figure supplement 2), suggesting a day-specific role for alpha2 receptors in sleep regulation. Bopindolol treatment results in reduction of night activity and increase in both day and night sleep (Figure 1—figure supplement 3). Similar to prazosin, clonidine and bopindolol may not activate and inhibit the all alpha2 and all beta receptor paralogs, respectively.

Our pharmacological manipulations do not perfectly recapitulate the *dbh* mutant phenotype. Rather, they serve as an independent set of evidence that noradrenergic signaling regulates sleep in larval zebrafish, as a way to demonstrate that in addition to genetic ablation of *dbh*, acute disruption of noradrenergic signaling also affects sleep, and as a way to investigate which adrenergic receptors mediate different aspects of the *dbh* phenotype. We have updated the manuscript to incorporate the new results, and we are grateful to the reviewers for encouraging us to expand our pharmacological studies.

*3) The result that adult* dbh-/- *fish have higher mortality related to normal doses of the anesthetic tricaine is interesting, yet out of place and unsupported by the data provided. You may consider omitting this finding, or explaining better its relevance.*

We have moved this description to the Methods section. While we agree that this observation is not critical for the paper, this information is nevertheless important for other labs who may wish to use these fish or who may be interested in this phenotype.

*4) With respect to the prazosin experiments, the investigators overstate the case for the effect not being dopamine-mediated. This would require measuring dopamine levels directly. It would be worthwhile to review (and to reference) the work from Ken Solt's lab showing that DA actions in VTA are important for arousal from anesthesia and to tone down the conclusion to “suggests the affect in not dopamine-mediated”, this may also be useful in the discussion surrounding*
Figure 3.

Please see comment 8.

*5) The paper claims in several places that “NE is required for Hcrt-induced arousal” (this quote is from the Abstract). This is not true. There is a major quantitative effect in the mutant (and see below) but as the arousal enhancement is not eliminated, this bold statement is incorrect*.

We have modified this statement to emphasize that Hcrt-induced arousal is reduced but not entirely eliminated in *dbh* mutant animals.

*6) It would have been good had they used the drug prazosin in combination with the Hcrt-induced arousal rather than just the mutant fish alone. NE elimination need not be the only effect of the KO, especially as it is systemic and all through development. For example, there could be compensatory changes in the endocrine system*. *Replication with the drug would increase confidence in the conclusion.*

In accordance with the reviewers’ suggestion, we treated *Tg(hsp:Hcrt+/-)* larvae with either vehicle or Prazosin. Hcrt induced arousal following heat shock induced overexpression of Hcrt was inhibited in the presence of prazosin, similarly to what we observed in *dbh* mutants. We have added a new supplementary (Figure 4—figure supplement 3) to the manuscript.

*7) How long was the pretreatment in*
Figure 1
*(how fast-acting is the drug), was a dose-response curve done (where is 100 μM on this curve?), and was the drug used in the* dbh *KO fish? Was it without effect as one might predict if it is specific for alpha-adrenergic receptors?*

In our hands prazosin treatment reduces activity and increases sleep within 2 hours of treatment (data not shown). We typically add the drug to the behavioral plate during the 4^th^ day of development and record sleep and locomotion starting at the morning of day 5. We have added this information to the manuscript.

We have tested different concentrations of Prazosin. Although as low as 10 μM of prazosin seems to saturate the sleep response, we noticed that different clutches of larvae showed considerable variability. Thus, in our experiments we raised the concentration we use to 100 μM; at this concentration we see the robust behavioral phenotype described in this manuscript with no lethality. Concentrations above 100 μM were not used, as above this point solubility becomes limiting and we observe formation of precipitate. We have added a new supplementary figure (Figure 1—figure supplement 1) with the dose response curve (prazosin concentration vs. sleep).

Also, as suggested by the reviewers we verified that prazosin treatment does not affect the sleep/wake behavior of *dbh* mutants and added this result to the manuscript (Figure 2—figure supplement 2).

*8) The arousal threshold effects (*Figure 3*) are rather modest, and enigmatic. First in*
Figure 3*, the “increased response rate” (not a response rate, just an enhanced response) is probably just an independent effect as the authors suggest. However, this means that they should replot their data subtracting a fixed value from each mutant response, and then compare with the WT. There may be a relatively small residual effect. Even in*
Figure 3*, it is difficult to interpret the magnitude of the effect because it is not clear what the X-axis is (log base 2, base 10?). The fact that the drug and the mutant effects are similar (even if modest) appears to underscore the conclusion but still… And if the drug result (3B) is really more impressive than the mutant (1A), what does this do to the dopamine hypothesis? Perhaps the authors could consider that the excess sleep itself lowers the arousal threshold*. *(Too much sleep; sleep more lightly?). And the more sleep itself is no surprise. Is this possible?*

Encouraged by the reviewers’ suggestions we made significant improvements to the arousal assay concerning both the mechanical aspects of stimulus delivery and the analysis of collected data. Specifically:

A) The tapping stimulus is now delivered by two solenoid devices on either side of the 96 well plate which houses the larvae. This ensures more even distribution of the tapping stimulus and allows us to increase its maximum force.

B) We increased the maximum voltage we can deliver to the solenoids from 12 V to 20 V and thus increased the maximum force the solenoids can deliver to the plate.

C) We refined the scale of stimuli we can deliver. The initial design allowed for a total of 99 different power settings, whereas the new design allows for over 4000 different settings. This allows us to use in the same experiment very small values that elicit no appreciable response, as well as very strong stimuli.

D) Finally, in accordance to the suggestion by the reviewers, we now calculate a baseline response ratio by measuring the probability of fish moving 5 seconds prior to the actual stimulus (background movement). This modification makes Figure 3–figure supplement 1 (movement probability in absence of tapping) no longer necessary, so we have removed it.

These changes to the assay have significantly altered Figure 3. The phenotype we observed for *dbh* mutants with the original assay is recapitulated with a more obvious curve shift (Figure 3). However, we were unable to reproducibly identify any phenotype following prazosin treatment. We presume this is a result of achieving a more accurate stimulus-response curve by employing a wider range of stimuli (changes B and C above) as well as correcting for background movement (change D).

We also performed the arousal assay using the alpha2 agonist clonidine and the beta antagonist bopindolol. Clonidine did not affect the arousal threshold of treated animals, but bopindolol treatment phenocopied the reduced arousal threshold of *dbh* mutants. These results suggest that acute pharmacological inactivation of beta-adrenergic receptors, but not alpha1 or alpha2 receptors, reduces the arousal threshold of zebrafish larvae.

Inactivation of dopamine beta hydroxylase is predicted to increase dopamine levels within neurons that normally express *dbh,* since dopamine beta hydroxylase is the enzyme that converts dopamine to norepinephrine. Indeed, dopamine levels have been shown to be higher in *dbh* mice compared to sibling controls (68). However, pharmacological inactivation of alpha1 receptors with prazosin, activation of the inhibitory alpha2 receptors with clonidine, or inactivation of the beta receptors with bopindolol do not interfere with the conversion of dopamine to norepinephrine. The ability of bopindolol to recapitulate the reduction in arousal threshold seen in the *dbh* mutant suggests that this aspect of the mutant phenotype is a consequence of absence of beta-adrenergic receptor signaling. Furthermore, the inability of prazosin (which increases sleep during day and night) and clonidine (which increases sleep during the day) to recapitulate the arousal threshold phenotype suggests that the observed reduction in arousal threshold in *dbh* mutants is not simply a consequence of excess sleep.

In the original manuscript we also performed a tap assay in which mutant and drug-treated larvae were exposed to the same stimulus every 5 min. The rationale was to test whether a mutation or a drug treatment differentially affected the behavior of awake larvae (larvae that had moved within the last minute) or asleep larvae (no movement within the last minute). The reason the stimulus was presented every 5 min instead of 1 min was to ensure adequate numbers of asleep fish for each data point. However, this proposition becomes problematic if we expose larvae with very different arousal thresholds to the same stimulus. In other words, even though we present all larvae with a stimulus of the same absolute magnitude, larvae of different genotypes or treated with different drugs can experience a very different subjective stimulus, rendering comparisons among them nonsensical. This limitation became apparent with the improved tapping assay used in the revised manuscript. As a result, we have removed experiments using a 5 minute inter-trial-interval (panels C and D from Figure 3).

It is important to note that these pharmacological results do not preclude a role of excess dopamine release by the previously noradrenergic neurons of the locus coeruleus and/or medulla oblongata in reducing the arousal threshold. This hypothesis is supported by zebrafish studies implicating dopamine in arousal modulation (10; 48) as well as work by the laboratory of Ken Solt demonstrating that dopaminergic stimulation during anesthesia produces a robust arousal response (65). Furthermore, dopamine agonists have been shown to increase locomotion in zebrafish larvae (56; 39) and dopamine promotes locomotor development in zebrafish larvae ([Bibr bib41a]).

We have updated the manuscript to reflect the new modifications and results. We have also modified the figure to more clearly highlight the altered response in mutant and drug treated larvae, and to more clearly explain the x-axes.

We are grateful to the reviewers for the significant improvements this comment, as well as comment #2, brought to the manuscript. We believe that the assay we employ, the way we analyze the data and the way we interpret these data are significantly improved based on the reviewers’ suggestions.

*9) The role of the LC is also no surprise given what has been done in rodents (e.g.,*
[13]*). It is however of value to pinpoint the role of NE within the LC. Given that the effect is far from complete, it means that there are other arousal systems that probably lie downstream of Hcrt – as indicated in the literature. It would be good if the authors discussed some of these and the likelihood that they also play a role in fish*.

We appreciate the reviewers pointing out this omission from the Discussion section. We have expanded the discussion of other arousal systems that may lie downstream of Hcrt and the likelihood that they also play a role in fish.

*10) The authors spend considerable effort emphasizing the controversy in the sleep phenotype of mouse* dbh *knockouts. While it is true that two groups reported different sleep results for those mice, it is unclear whether this fish study truly resolves the controversy – it only provides supporting evidence that NA is wake-promoting in a phylogenetic sense and is consistent with analysis of the role of octopamine in* Drosophila *sleep. The evidence they provide for hypocretin promoting arousal partially via NA is a nice extension of a prior study done in mice by*
[13]*, which implicated LC neurons in the arousing effects of hypocretin*.

We have modified our statements as suggested to emphasize that our results are consistent with the demonstrated role of octopamine in *Drosophila* sleep and with mammalian data suggesting that NA is wake-promoting.